# Score-based generative modeling through anisotropic SPDEs

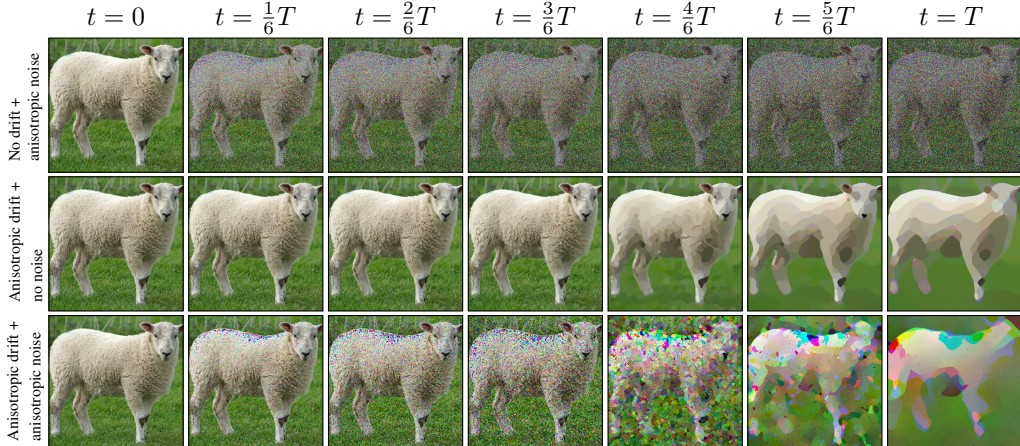

Figure 1: We visualize forward processes within our anisotropic diffusion framework. From top to bottom: (1) isotropic drift with anisotropic noise, (2) anisotropic drift without noise, and (3) both anisotropic drift and noise. Different anisotropy settings influence the preservation of geometric structures during the forward process, thereby affecting their reconstructability during generative sampling.

## ABSTRACT

Score-based generative modeling (SBGM) has achieved state-of-the-art performance in image generation, with the quality of generated images highly dependent on the design of the forward (diffusion) process. Among these, models based on stochastic differential equations (SDEs) have proven particularly effective. While traditional methods aim to progressively destroy all image information to enable reconstruction from pure noise, we introduce a novel class of anisotropic stochastic partial differential equations (SPDEs) that preserve the geometric structure of the data throughout the transformation. These SPDEs consist of a drift term that enforces deterministic destruction via structured smoothing, and a diffusion coefficient that enables random destruction through noise injection. Both components are governed by anisotropy coefficients, enabling controlled, direction-dependent information degradation. This framework provides the theoretical foundation for a novel anisotropic SBGM. Due to geometry-aware degradation, the data generation process can exploit residual geometric cues, leading to improved fidelity in image reconstruction. We empirically validate this improvement in a proof-of-concept implementation on unconditional image generation, showing that anisotropic diffusion can achieve superior image quality metrics.

## 1 INTRODUCTION

Diffusion-based generative models (Song & Ermon, 2019; Ho et al., 2020) have gained significant traction due to their remarkable ability for high-quality image synthesis in both conditional and unconditional settings. Their impressive performance in modeling high-dimensional datasets of considerable size has led to their adoption across various other fields, such as video synthesis (Xing et al., 2024), speech and audio synthesis (Kong et al., 2020; Huang et al., 2022), medical imaging (Song et al., 2021a; Kazerouni et al., 2023), and molecular design (Weiss et al., 2023; Schneuing et al., 2024).

Broadly, diffusion-based generative models can be categorized into two classes. The first class comprises SBGMs, which are based on learning the gradient of the log-density (the *score*) of the data transformation.

These models are further divided into two subcategories: The first category, ordinary score matching (OSM) (Song & Ermon, 2019), directly estimates the score of the data distribution itself. The second category, Denoising Score Matching (DSM), learns instead the score of perturbed data, which includes methods such as Score Matching with Langevin Dynamics (SMLD) (Song & Ermon, 2019), Denoising Diffusion Probabilistic Models (DDPM) (Ho et al., 2020), and SDE-driven score-based diffusion models (SDE-driven SBGMs) (Song et al., 2021c). In this paradigm, a noise perturbation is applied to the data, and the model learns to approximate the score of this transformed distribution at each noise level. This learned score can then be used within various sampling algorithms to generate new data points.

The second class of diffusion-based generative models follows an alternative approach, as introduced by Sohl-Dickstein et al. (2015). Rather than explicitly estimating the score function, these models define a latent variable hierarchy and optimize a variational bound to maximize the likelihood of the data. While these methods share conceptual similarities with DDPM, their formulation does not rely on score estimation and instead focuses on parameterizing the forward and reverse processes explicitly.

In this work, we adopt the approach of the former class. More precisely, we build up on and extend the framework of SDE-driven SBGMs. These SDE-driven models provide a powerful mathematical framework with fine-grained control and flexibility when it comes to designing the generative process. The framework we introduce allows us to consider anisotropic diffusion processes that are aware of the geometrical structures present in the underlying data. Inspired by promising results from previous work (Yu et al., 2023; Vandersanden et al., 2024) on anisotropic diffusion processes, our hope is that our framework enables the design of better performing generative diffusion models.

Our main contributions are the following:

- **Anisotropic Diffusion Framework**: We introduce a novel anisotropic diffusion framework that respects geometric structures during data destruction, facilitating the resemblance of geometric features in the generative sampling process (Section 4).
- **Empirical Validation**: We demonstrate the potential of our theoretical framework through a numerical study of a proof-of-concept implementation on unconditional image generation (Section 5).

In addition, we provide a unifying formulation of SBGM that is a straightforward generalization of existing work (Lim et al., 2023; 2024) (Section 3), extending it to encompass our class of anisotropic diffusion processes as well. This serves primarily as a by-product: it allows us to describe training and generative sampling in a common framework that subsumes both existing approaches and our anisotropic diffusion framework, thereby avoiding a separate, framework-specific presentation.

## 2 RELATED WORK

**Conceptually related models**  Song et al. (2021c) first used time SDEs for SBGMs and showed how existing diffusion models can be unified by an SDE framework. However, they only considered linear SDEs with spatially independent diffusion coefficients.

Rissanen et al. (2023) considered a stochastic heat equation with isotropic noise, which is effectively destroying the data by blurring up to complete dissipation. This is in contrast to earlier approaches that typically destroyed data into pure noise. Hoogeboom & Salimans (2022) extended this idea by introducing a temporally increasing isotropic noise term, further refining the blurring process over time.

**State-of-the-art models**  Lipman et al. (2022) propose Flow Matching (FM), a simulation-free training method for continuous normalizing flows (CNFs) that regresses vector fields along predefined probability paths. FM enables training CNFs with more efficient paths such as optimal transport interpolations, yielding faster sampling and superior sample quality. We refer to Section B for a discussion on introducing anisotropy in FM.

Zhou et al. (2023) introduce Denoising Diffusion Bridge Models (DDBM), which generalize diffusion models to map between arbitrary endpoint distributions using learned diffusion bridges. This framework unifies generative modeling paradigms and enables tasks like image translation, achieving strong performance while remaining competitive with state-of-the-art models in standard generation settings. Conditional generative modeling through anisotropic diffusion bridges is part of future work; we elaborate on that in Section C.

**Anisotropic models**  Several studies (Voleti et al., 2022; Yu et al., 2023; Vandersanden et al., 2024) explored the role of anisotropic noise in diffusion models. Vandersanden et al. (2024) propose a structure-aware anisotropic diffusion process that preserves edges longer, improving sample quality, particularly in shape-oriented generative tasks.

Our approach shares similarities with this work, as the anisotropic SPDE we introduce is also guided by structural image content in both the drift and diffusion terms. However, our method differs in that it models a

genuinely anisotropic and nonlinear diffusion process, where both the drift and diffusion coefficients evolve dynamically based on the current state, rather than being fixed by the initial state.

**SPDE-based models**  Lim et al. (2023; 2024) also consider generative modeling using SPDEs. The parabolic SPDE studied in Lim et al. (2024) is restricted to spatially-independent diffusion coefficients. This limitation prevents the modeling of spatially varying or anisotropic effects in the forward process, and therefore represents a more constrained setting than our framework that allows for general anisotropic diffusion.

## 3  Unified formulation of SBGM

In this section, we establish a unified formulation of SBGM that encompasses existing approaches on SBGM, including OSM, SMLD, DDPM and SDE-driven SBGMs, as well as our anisotropic diffusion framework, which we propose in Section 4. Our unifying formulation is a straightforward generalization of existing work (Lim et al., 2023; 2024), extending it to encompass our class of anisotropic diffusion processes as well.

This unifying perspective allows training and data generation to be described in a common, high-level fashion that applies uniformly across all existing SBGMs — including ours. In this way, our framework can be presented within the same generic algorithmic structure as prior work, rather than through bespoke descriptions that might suggest a fundamental methodological departure.

**Overview**  Generative modeling operates as a two-pass procedure: In the first (*forward*) pass, considered in Section 3.1, the information in the data is systematically destroyed to a certain extent. During this pass, the score (i.e. the log-density) of the forward process is learned (by a neural network) — as explained in Section 3.2.

In the *backward* pass, the destroyed data is stochastically reconstructed, resulting in new, previously unseen samples resembling the original data. Because of this dual perspective, the transformation $(U_t)_{t \in \overline{I}}$ is called the *forward process*, while its time reversal

$$\overline{U}_t := U_{T-t} \quad \text{for } t \in \overline{I}. \tag{1}$$

is referred to as the *backward process*. In Section 3.4, we explain how it can be used for generative sampling.

### 3.1  Forward pass (data destruction)

In generative modeling, the goal is to learn a data distribution $\mu$ and generate new samples that closely resemble the data. Conceptually, the data distribution $\mu$ is a probability measure on $\mathbb{R}^D$, where $D$ is a finite (index) set. In practice, $\mu$ is unknown and only implicitly given by a dataset, where we assume that this dataset is an independent and identically $\mu$-distributed sequence.

More specifically, in *score-based* generative modeling, the (complex) data is mapped by a (stochastic) transform to a simpler representation, and the *score* (i.e., the log-density) of this transformation is learned. Conceptually, the information in the data is progressively destroyed until fully degraded, after which the objective is to generate new data resembling the original distribution. From an analytical perspective, the data is smoothed over time, progressively simplifying the learning task — a practice referred to as *regularization by noise*.

Historically, various types of transformations have been explored in SBGM. To describe all of them, including our approach proposed in Section 4, in a unified manner, let $\overline{I} \subseteq [0, \infty)$ denote the time index set of the transformation, and let $(U_t) t \in \overline{I}$ represent the transformation itself. Formally, $(U_t)_{t \in \overline{I}}$ is an $\mathbb{R}^D$-valued (stochastic) process with initial distribution $\mu$, i.e., it begins by sampling its initial state from the data distribution $\mu$.

### 3.2  Learning the score function

The score of the forward process provides the necessary information for reconstructing samples in the generative sampling process. Without a score, the reconstruction would be purely deterministic, meaning that the generative sampling process would have to exactly invert the forward transformation. However, exact inversion is often impossible, as the forward process is designed to progressively degrade information in a way that cannot be deterministically undone.

By incorporating stochasticity into the forward dynamics, we ensure that the degradation is probabilistic, rendering the reverse process well-posed in a statistical sense. We approximate the score using a neural network trained during the forward pass (Section 3.1) and leverage it to guide generative sampling, the generative sampling process, as described in Section 3.4.

That is, for SBGM, we need to make sure that the score of the forward process actually exists. Consequentially, we assume that $U_t$ has a positive differentiable density $p_t$ with respect to the $D$-dimensional Lebesgue measure

for all $t \in I$. The *score* of the transformation at time $t \in I$ is now defined to be

$$s(t, \cdot) := \nabla \ln p_t. \tag{2}$$

The goal of SBGM is now to train a (time-dependent) score-based model to find an approximation of $s$. To this end, we need a suitable metric measuring the distance between any given approximation $\tilde{s}$ to the true score $s$. Existing approaches have consistently relied on an $L^2$-norm for that purpose. We now derive a *loss measure*, unifying OSM, SMLD, DDPM and SDE-driven SBGMs, with respect to which this $L^2$-norm is defined.

The time domain $\overline{I}$ of the forward process can either be discrete or continuous. We assume that $0 \in \overline{I}$ and

$$T := \sup \overline{I} \in [0, \infty) \setminus I \tag{3}$$

for a smaller subset $I \subset \overline{I}$, which is actually used during the learning process as we will describe now. Introducing a *loss measure*

$$\eta(A \times B) := \int_A \mathcal{U}_I(\mathrm{d}t)\zeta(t)\,\mathrm{P}[U_t \in B] \tag{4}$$

for measurable $(A, B) \subseteq I \times \mathbb{R}^d$, the typical *loss function* $L$ is the $L^2(\eta)$-distance to the actual score $s$; i.e.

$$L(\tilde{s}) := \|\tilde{s} - s\|_{L^2(\eta)}^2 = \int \mathcal{U}_I(\mathrm{d}t)\zeta(t)\,\mathrm{E}\big[\|(\tilde{s} - s)(U_t)\|^2\big] \tag{5}$$

for measurable $\tilde{s} : I \times \mathbb{R}^d \to \mathbb{R}^d$, where $\mathcal{U}_I$ denotes the uniform distribution on $I$. In this definition, $\zeta : I \to [0, \infty)$ is a *weighting* function allowing us to put more importance on certain parts of the transformation.

## 3.3 UNIFYING PRIOR WORK

As apparent from the definition, the loss is not considered over the entire time domain $\overline{I}$ of the transformation, but only over the smaller subset $I$. With this description, we capture all of the previous works. They differ in how they handle the forward process and score estimation. With our unified formulation, we got

○ $I = \{0\}$ and $\overline{I} = I \uplus \{1\} = \{0, 1\}$ in OSM;

○ $I = \{1, \ldots, k-1\}$ and $\overline{I} = \{0\} \uplus I \uplus \{k\} = \{0, \ldots, k\}$
for some $k \in \mathbb{N}$ with $k \geq 2$ in SMLD and DDPM; and

○ $I = [t_0, T)$ for some $0 < t_0 < T < \infty$ and $\overline{I} = [t_0, T]$ in SDE-driven SBGMs.

That is, in OSM, the data from $\mu$ is not actually transformed, but remains unchanged. Instead, the score $s(0, \cdot)$ of the data distribution $\mu$ is tried to be learned directly. To generate samples, we initialize from a *prior* distribution, which corresponds to the distribution of $U_T$ — the state of the forward process at the final time $T$. In this setting, the transformation by the forward process is effectively performed in a single step by drawing from the prior, without dependence on an initial dataset sample. Thus, the index set of the transformation is given by $\overline{I} = \{0, 1\}$.

In contrast, SMLD, DDPM, and SDE-driven SBGMs do not aim to learn the score of the data distribution directly. Instead, they transform dataset samples through a nontrivial forward process, producing representations at multiple (potentially infinitely many) *noise*, or — more generally — *diffusivity levels*, and not at the initial stage corresponding to the data distribution directly. These models then learn the score of the transformed distributions at each diffusivity level.

In SMLD and DDPM, the forward process operates at a discrete set of diffusivity levels, indexed by $\overline{I} = \{0, \ldots, k\}$. In contrast, in SDE-driven SBGMs, the forward process is continuous, resulting in a continuous range of diffusivity levels with index set $\overline{I} = \{0\} \cup [t_0, \infty)$ for some $t_0 > 0$, where score learning begins at $t_0$.

**Prior sampling** The distribution of $U_T$ is being referred to as the *prior* distribution of the backward, data generation pass. A sample from the prior distribution $U_T$ is formally obtained by initializing the forward process $(U_t)_{t \in \overline{I}}$ with a random sample from the data distribution $\mu$ and simulating it up to the terminal time $T$. Depending on the complexity of the prior, this may be the only practical sampling procedure; for example, this is the approach taken in Rissanen et al. (2023).

However, in OSM, the prior distribution is completely decoupled from the initial data distribution $\mu$ from which $U_0$ is initialized. In fact, in traditional methods, the prior distribution is typically (approximately) Gaussian.

## 3.4 BACKWARD PASS (DATA GENERATION)

Once we have trained a time-dependent score-based model to approximately minimize $L$, we can generate new, previously unseen data that resembles the training data distribution $\mu$.

This generation process follows an iterative sampling scheme. It begins with a draw from the prior distribution. At each iteration, the scheme applies an optional *predictor* step, followed by an optional *corrector* step.

- The predictor step, if applied, propagates the sample backward in time by simulating a step of the backward process. This step may incorporate information from previous samples, which is particularly beneficial when the backward process has a Markov property.
- The corrector step, if applied, refines the sample using unadjusted Langevin algorithm (ULA), treating the sample from the previous iteration as the initial state and targeting the distribution at the corresponding time step. This is feasible because ULA only requires the gradient of the log-density, which can be estimated using the learned score function. However, using the *Metropolis-adjusted* ULA would necessitate additional density estimation techniques.

The full procedure is summarized in Algorithm 3.1. Depending on the specific score-based generative framework, only the predictor step (e.g., DDPM, SDE-driven SBGMs), only the corrector step (e.g., OSM, SMLD), or both (as in SDE-driven SBGMs) may be used. However, at least one of these steps must be applied.

How sampling in the predictor step can be performed, completely depends on the complexity of the forward process. If it has a Markov property or is given as the solution of an SDE, as it is the case for our forward process introduced in Section 4, special sampling techniques are available. For details, we refer to Section D.

---

**Algorithm 3.1** Generative sampling process

---

**Output:** Sample $u_0$ from the data distribution $\mu$
1: **if** $I$ is continuous
2:      Choose $k \in \mathbb{N}$ and (strictly) increasing $t_0, \dots, t_{k-1} \in I$;
3: **else**
4:      Enumerate $I = \{t_0 < \cdots < t_{k-1}\}$, where $k := |I|$;
5: Sample $u_k$ from the prior distribution $U_T$;
6: **for** $i = k, \dots, 1$
7:      Sample $u_{i-1}$ from $U_{t_{i-1}}$ given $u_i, \dots, u_k$;
     {Optional predictor step}
8:      Correct $u_{i-1}$ by applying ULA with initial state $u_{i-1}$ and target density $p_{t_{i-1}}$;
     {Optional corrector step}

---

With Algorithm 3.1, we capture all of the existing work. In OSM, Algorithm 3.1 employs only the corrector step, using ULA iterations with the learned score and the prior sample as the starting point. In SMLD, Algorithm 3.1 also applies only the corrector step, meaning ULA iterations are performed sequentially, descending through diffusivity levels from an initial prior sample.

In contrast, in DDPM and SDE-driven SBGMs, the predictor step in Algorithm 3.1 is used, where samples are generated by sequentially following the dynamics of the learned backward process, which is a Markov process in those frameworks.

SDE-driven SBGMs can also incorporate a corrector step. Unlike DDPM, where exact sampling from the backward process is feasible, SDE-driven SBGMs only allow approximate sampling due to continuous-time dynamics. Consequently, ULA can be interleaved with predictor steps to correct bias introduced by approximation errors.

## 4 ANISOTROPIC DIFFUSION FRAMEWORK

In this section, we describe our anisotropic diffusion framework. We extend conventional SDE-based approaches by formulating the forward process as the solution of an SPDE. This formulation naturally incorporates spatial derivatives, enabling structured and anisotropic transformations of images.

Unlike conventional SBGMs that operate directly on the discrete pixel grid, we formally treat each color channel of the image as a function $\Lambda \to \mathbb{R}$ evolving over a continuous spatial domain $\Lambda$.

Mathematically, we define $\Lambda$ as a bounded open subset of $\mathbb{R}^d$, where $d \in \mathbb{N}$ represents the spatial dimension. In our application, images are naturally two-dimensional, so we have $d = 2$. A common choice is to model the image space $\Lambda$ as $(0, 1)^2$ for normalized coordinates or as $\Lambda = (0, \texttt{width}) \times (0, \texttt{height})$ in the physical pixel space, where $\texttt{width}$ and $\texttt{height}$ denote the dimensions of the image in pixels.

### 4.1 ANISOTROPIC FORWARD PROCESS: THE THEORETICAL MODEL

We now define the specific SPDE our forward process will satisfy. The equation describes how information is progressively diffused in an anisotropic manner, ensuring a structured degradation of images and enhanced reconstruction capability of geometric features in the generative sampling process (Algorithm 3.1).

To this end, we propose to model the forward process $(U_t)_{t \in \overline{I}}$ as the *formal* solution of

$$\mathrm{d}U_t = b(t, U_t)\,\mathrm{d}t + \sigma(t, U_t)\,\mathrm{d}W_t \quad \text{for all } t \in \overline{I}, \tag{6}$$

where

$$b(t, u) := \nabla \cdot g_1(t, \nabla u)\nabla u \tag{7}$$

for $(t, u) \in \overline{I} \times H^2(\Lambda)$,

$$\sigma(t, u)v := g_2(t, \nabla u)v \tag{8}$$

for $(t, u) \in \overline{I} \times H^1(\Lambda)$ and $v \in Q^{1/2}L^2(\Lambda)$,

$$g_i(t, x) := \frac{\alpha_i(t)}{\sqrt{1 + \left\| \dfrac{x}{\lambda_i(t)} \right\|^2}} \tag{9}$$

for $(t, x) \in I \times \mathbb{R}^d$, $\alpha_i : \overline{I} \to [0, \infty)$ and $\lambda_i : \overline{I} \to (0, \infty]$ are continuous and nondecreasing and $(W_t)_{t \in \overline{I}}$ is a $Q$-Wiener process (Da Prato & Zabczyk, 2014, Definition 4.2) with

$$(Qf)(x) := \int_\Lambda q(x, y)f(y)\,\mathrm{d}y \quad \text{for } x \in \Lambda \tag{10}$$

for $f \in L^2(\Lambda)$ and, for some $\ell \in [0, \infty)$,

$$q(x, y) := \exp\left( -\frac{\|x - y\|}{\ell} \right) \quad \text{for } x, y \in \mathbb{R}^d. \tag{11}$$

As usual, $H^r(\Lambda)$ denotes the Hilbert Sobolev space (Renardy & Rogers, 2004, Chapter 7) of order $r \in \mathbb{N}_0$.

The reader, which is interested in, but unfamiliar with, stochastic analysis can just think about an $Q$-Wiener process as a generalization of a standard Wiener process (or Brownian motion) whose covariance operator is given by $Q$ and refer to (Da Prato & Zabczyk, 2014; Lord et al., 2014) for details. Especially, integral operators of the form (10) are considered in (Lord et al., 2014, Definition 1.64). A discussion on the *Cameron-Martin space* $Q^{1/2}L^2(\Lambda)$ can be found in Section F.

(6) is the natural stochastic generalization of the deterministic Perona—Malik diffusion Perona & Malik (1990). In Section A, we provide design guidelines by detailing how the individual ingredients — the diffusivity coefficients $\alpha_k$ (Section A.1.1), intensity coefficient (Section A.1.2), anisotropy coefficient $\lambda_i$ (Section A.1.3) and *correlation length* $\ell$ (Section E.2) — of our anisotropic diffusion framework control the image transformations described by our forward process. A classification of the SPDE types arising from different parameter choices is provided in Section G.

## 4.2 PRACTICAL FORWARD AND BACKWARD PROCESSES

From a strictly mathematical perspective, ensuring correctness of the generative modeling methodology requires that the backward process is the *exact* time-reversal of the forward process and that both forward and backward processes are *exact* simulatable. However, for a complex SPDE like (6), exact simulation is impossible. Both spatial and temporal discretization must be performed to obtain a practically simulatable process.

Spatial discretization, whether by Galerkin methods (as in (Lim et al., 2024)) or finite differences (as in our numerical scheme), inevitably leads to a projection onto a finite-dimensional SDE of the form

$$\mathrm{d}\tilde{U}_t = \tilde{b}\left(t, \tilde{U}_t\right)\mathrm{d}t + \tilde{\sigma}\left(t, \tilde{U}_t\right)\mathrm{d}\tilde{W}_t \quad \text{for all } t \in \overline{I}, \tag{12}$$

where $\tilde{b} : \mathbb{R}^D \to \mathbb{R}^D$, $\tilde{\sigma} : \mathbb{R}^D \to \mathbb{R}^{D \times D}$, and $(\tilde{W}_t)_{t \geq 0}$ is a $D$-dimensional Brownian motion for some $D \in \mathbb{N}$.

This procedure unavoidably introduces approximation error, which is further compounded by the subsequent temporal discretization. In special cases — such as the parabolic SPDEs with additive noise considered in Lim et al. (2024) — this may be acceptable. Nevertheless, if one uses the time-reversal of the infinite-dimensional SPDE as the backward process, a mismatch arises between the processes that can actually be simulated and the theoretical foundation of their use, and exact correctness of the backward process is no longer guaranteed.

In our framework, we consider a substantially more complex SPDE — with gradient-dependent nonlinear drift and gradient-dependent multiplicative noise — than in previous SPDE-based SBGMs (e.g., Lim et al. (2024)). In our case, no fixed eigenbasis diagonalizes the drift operator or the noise, and modal decoupling is unavailable.

We therefore adopt a different strategy: we first apply spatial discretization, thereby defining the actual forward process directly as the finite-dimensional SDE (12). The backward process is then taken as the time-reversal of this SDE, ensuring that both forward and backward dynamics are defined at the same level of approximation.

As a result, the only simulation error arises from temporal discretization, rather than from a combination of spatial *and* temporal discretizations.

The concrete numerical scheme leading to the SDE (12) used in our experiments in Section 5 is described in Section H. Theoretical existence of a solution to (12) is verified by Pascucci (2011, Theorem 9.11). That the corresponding backward process also satisfies an SDE follows from classical results (Haussmann & Pardoux, 1986; Anderson, 1982). For the explicit form of this SDE we refer to Section D.

## 4.3 RESIDUAL DEPENDENCE ON THE INITIAL STATE

Introducing anisotropy inherently induces a residual dependence on the initial state. Our framework contains many user-definable parameters. For meaningful use in a generative modeling context, the modeling process that determines these parameters should ensure that the information contained in the initial state is almost entirely destroyed by the terminal time $T$. Conceptually, anisotropy should only serve to prolong the preservation of certain structures (such as edges) to facilitate the reconstruction of geometric features during data generation.

For specific parameter choices in our framework, the forward process reduces to an Ornstein–Uhlenbeck process (as in the instance described in Section 5.3), or it can be designed such that the distribution at the terminal time $T$ is close to a known Gaussian distribution (as in the instance described in Section 5.2). In such cases, the prior distribution can be replaced by this closed-form distribution when performing Algorithm 3.1.

Whether the prior distribution admits a closed form (or a tractable approximation) depends on the chosen parameters. If not, a sample from the prior can be generated by simulating the forward process up to the terminal time. In either case, mathematical correctness is ensured; see our discussion in Section 3.3.

## 5 NUMERICAL STUDY

Intuitively, the anisotropy introduced by our anisotropic diffusion framework helps preserve structural information over longer time scales during the forward transformation. This, in turn, facilitates learning and reconstruction of geometric features during data generation.

In this section, we empirically validate this intuition through a numerical study on unconditional image generation. We compare our framework against three baselines: Rissanen et al. (2023), the *variance exploding SDE (VESDE)* from Song et al. (2021c), and the *flow matching / optimal transport* method of Lipman et al. (2022), which is state of the art at the time of writing.

We focus on two specific instances of our framework: an isotropic version (*Ours (isotropic)* described in Section 5.3) and an anisotropic version (*Ours (anisotropic)* described in Section 5.2) of a stochastic heat equation. The isotropic variant is included primarily for educational purposes, as it demonstrates that the model equation of Rissanen et al. (2023) arises as a special case of our framework.

Both Rissanen et al. (2023) and Song et al. (2021c) provide especially relevant baselines, since they are likewise based on S(P)DEs. In contrast to our approach, however, their drift and diffusion coefficients are isotropic. Conceptually, the only difference between their methods and *Ours (anisotropic)* is the introduction of anisotropy. This design choice allows us to attribute observed improvements directly to anisotropy.

We now describe the specific instances of our framework considered in the numerical study (Sections 5.1 to 5.3), followed by a detailed account of the experiments in Section 5.4.

### 5.1 PURE ISOTROPIC NOISE (SONG ET AL. (2021C))

$$dU_t = \alpha_2(t) \, dW_t \quad \text{for all } t \in \overline{I}. \tag{13}$$

The SPDE formulated in (13) defines a Gaussian process, of which the *VESDE* considered in Song et al. (2021c) is a specific instance. It has a vanishing drift term, $b = 0$, and an isotropic diffusion coefficient, $\sigma$. Intuitively, this corresponds to a process in which an increasing amount of noise is added to the data over time.

### 5.2 ANISOTROPIC STOCHASTIC HEAT EQUATION WITH ISOTROPIC NOISE (OURS (ANISOTROPIC))

$$dU_t = \nabla \cdot \frac{\alpha_1(t)}{\sqrt{1 + \left\| \frac{\nabla U_t}{\lambda_1} \right\|^2}} \nabla U_t \, dt + \alpha_2(t) \, dW_t \quad \text{for all } t \in \overline{I} \tag{14}$$

The SPDE in (14) is a genuinely anisotropic instance of our general anisotropic diffusion framework (6), where the drift gradually transitions from anisotropy to isotropy while the diffusion coefficient remains isotropic. We consider *geometric* transitions of the form

$$\alpha_i(t) := \alpha_i^{\min}\left(\frac{\alpha_i^{\max}}{\alpha_i^{\min}}\right)^{\frac{t}{T}} \quad \text{for } t \in \overline{I} \tag{15}$$

for $0 < \alpha_i^{\min} < \alpha_i^{\max}$ (other common transitions are shown in Figure 5). Specifically, $\alpha_1$ increases geometrically from $\alpha_1^{\min} = 0.5$ to $\alpha_1^{\max} = 2 \cdot \texttt{image\_size}$, while the anisotropy coefficient $\lambda_1$ ensures a slow transition from anisotropy to isotropy via

$$\lambda_1(t) := \lambda_1^{\min}\frac{e^{kT} - 1}{e^{k(T-t)} - 1} \quad \text{for } t \in \overline{I} \tag{16}$$

with $\lambda_1^{\min} = 0.025$ and $k = 1/2$ (see Figure 5 for a visualization). The intensity coefficient $\alpha_2$ also increases geometrically, from $\alpha_2^{\min} = 0.01$ to $\alpha_2^{\max} = 2.0$. With $\lambda_2 \equiv \infty$, the diffusion coefficient remains spatially isotropic throughout. We set the noise correlation length to $\ell = 0$, leading to a *cylindrical* Wiener process $(W_t)_{t \in \overline{I}}$ and hence spatially white noise. The corresponding forward and backward processes are visualized in Figure 8.

Since $\lambda_1(t) \to \infty$ as $t \to T$, the numerical simulation (12) of the SPDE (14) is, at least approximatively, conditionally Gaussian given the initial state. Hence, prior sampling can be performed from a closed-form (Gaussian) distribution in the implementation.

### 5.3 STOCHASTIC HEAT EQUATION WITH ISOTROPIC NOISE (RISSANEN ET AL. (2023) AND OURS (ISOTROPIC))

$$dU_t = \alpha_1(t)\Delta U_t \, dt + \alpha_2(t) \, dW_t \quad \text{for all } t \in \overline{I}. \tag{17}$$

The SPDE in (17) has an isotropic drift $b$ and a *small-scale* isotropic diffusion coefficient $\sigma$. It closely resembles the forward process considered by Rissanen et al. (2023). Intuitively, the data is smoothed over time, and a small amount of noise is injected to make the forward process stochastic, which is required to ensure that the reverse process is well-posed (see Section 3.2).

For $\alpha_1$, we again use a geometric transition (15) with $\alpha_1^{\min} = 0.5$ and $\alpha_1^{\max} = 2 \cdot \texttt{image\_size}$. A minor difference from Rissanen et al. (2023) is that we do not keep the intensity coefficient $\alpha_2$ constant; instead, it increases slightly over time (while remaining small) under a geometric transition equation 15 with $\alpha_2^{\min} = 0.01$ and $\alpha_2^{\max} = 0.5$. Formally, the anisotropy coefficients are fixed as $\lambda_1 = \lambda_2 \equiv \infty$. The noise correlation length is again set to $\ell = 0$, so we also work with a cylindrical Wiener process $(W_t)_{t \in \overline{I}}$ here.

With the parameter choices described above, the numerical simulation (12) of the SPDE (17) is conditionally Gaussian given the initial state. Consequently, prior sampling can be carried out from a closed-form (Gaussian) distribution in the implementation.

### 5.4 EXERPIMENTS

In our experiments, our methods *Ours (anisotropic)* and *Ours (isotropic)* used both the predictor and corrector steps of Algorithm 3.1. Each corrector step consists of a single ULA iteration. We simulated up to a final time of $T = 2$ with a numerical step size of $0.001$ in the predictor step, resulting in 1999 score function evaluations.

**Limitations** The flexibility of our framework opens the door to exploring a wide range of anisotropic forward processes beyond *Ours (anisotropic)*, which may further enhance quality. However, due to hardware resource limitations, this study is limited to the single parameter configuration given by *Ours (anisotropic)*, a small number of datasets listed below, and the restricted set of baseline methods mentioned above.

**Test datasets** We trained all generative models on the CIFAR-10 (Krizhevsky et al., 2009), CelebA (Liu et al., 2015), ImageNet2012 (Russakovsky et al., 2015), and LSUN/church_outdoor (Yu et al., 2015) datasets.

**Evaluation** We assess the quality of generated samples using standard metrics in generative image modeling: the Inception Score (IS) (Salimans et al., 2016), Fréchet Inception Distance (FID) (Heusel et al., 2017), and Kernel Inception Distance (KID) (Bińkowski et al., 2021). To ensure consistent evaluation, we used the implementation (Song et al., 2021b) provided by Song et al. (2021c) and regenerated all samples — including those for baseline methods — to compute these metrics. Because metric implementations vary slightly across toolsets, our reported values may differ from those originally published. This makes it particularly important that all models are evaluated under identical conditions.

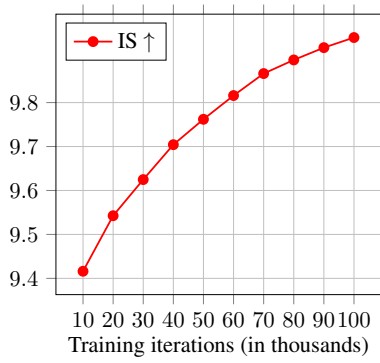 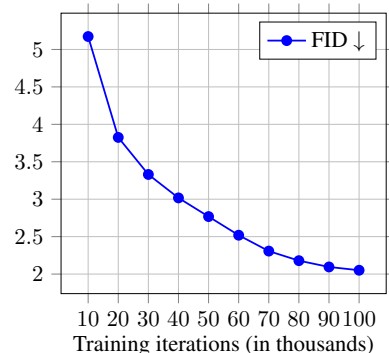

Figure 2: Evaluation metrics on CIFAR-10 between 10k and 100k training iterations with *Ours (anisotropic)*. We employed a fine-tuning strategy, initializing training from the checkpoint of Song et al. (2021c). For reference, applying our sampling procedure directly to the Song et al. (2021c) checkpoint —— without additional training —— results in an IS of 1, FID of 678.258, and KID of 0.872. After 100k training iterations, these values improve substantially to an IS of 9.978, FID of 2.041, and KID of 6.411.

**Quantitative comparison**    In Table 1, we report the evaluation metrics of samples generated by the different methods on the test datasets. All values reported for *Ours (isotropic)* and *Ours (anisotropic)* are based on training from scratch for 200,000 steps. Since no checkpoints are provided by the authors, we retrained the method of Lipman et al. (2022) on all datasets using the official configuration from their codebase for the same number of steps. The same procedure was applied to Rissanen et al. (2023), although we restricted evaluation to CIFAR-10, given the method's already non-competitive performance on this dataset. For Song et al. (2021c), we trained CelebA and ImageNet2012 from scratch, while for CIFAR-10 and LSUN/church_outdoor we relied on the official checkpoints released by the authors.

Figure 2 illustrates how quickly (i.e., after how few training iterations) *Ours (anisotropic)* improves the evaluation metrics when initialized from a model pretrained with Song et al. (2021c). Notably, according to the original authors, continuing training with their own method did not yield further metric improvements.

| | IS ↑ | FID ↓ | KID ↓ |
|---|---|---|---|
| **Ours (anisotropic)** | **10.2** / **3.1** / **13.5** / **3.6** | **2.0** / 2.4 / **19.1** / 5.9 | **6.1e-4** / 1.7e-3 / **1.9e-2** / **4.6e-3** |
| Lipman et al. (2022) | 9.2 / 2.4 / 10.5 / - | **2.0** / **2.3** / 26.8 / - | 7.1e-4 / **1.4e-3** / 3.4e-2 / - |
| Song et al. (2021c) | 9.8 / 2.5 / 12.3 / 2.5 | 7.1 / 3.7 / 24.0 / 16.7 | 6.6e-4 / 2.6e-3 / 2.5e-2 / 1.2e-2 |
| Vandersanden et al. (2024) | 7.1 / 2.7 / - / 3.4 | 28.7 / 12.0 / - / 49.1 | 2.2e-2 / 8.4e-3 / - / 4.3e-2 |

Table 1: Each column within the IS, FID, and KID metrics corresponds, in order, to results on CIFAR-10 (32×32), CelebA (64x64), ImageNet2012 (64x64), and LSUN/church_outdoor (256x256). Higher IS is better, while lower FID and KID are better. On CIFAR-10, we also compared *Ours (isotropic)* with Rissanen et al. (2023). *Ours (isotropic)* achieved metrics of 8.8 / 19.6 / 1.6e-02, while Rissanen et al. (2023) produced the worse metrics 5.9 / 84.3 / 7.2e-2. *Ours (isotropic)* and *Ours (anisotropic)* refer to the isotropic and anisotropic stochastic heat equation with isotropic noise described in Section 5.3 and Section 5.2, respectively.

**Hardware resources**    All experiments were conducted on a server equipped with 8× NVIDIA Tesla H100 NVL GPUs (94 GB HBM3 each, PCIe 5.0) and 2× AMD EPYC 9554 CPUs (64 cores / 128 threads each, 3.1–3.75 GHz, Genoa microarchitecture, 256 MB L3 cache).

**Hyperparameters and architecture**    For the implementation of our framework, we adopted the NCSN++ (continuous) network architecture from Song et al. (2021c). The training parameters —— in particular, a learning rate of 2e-4 using the Adam optimizer (Kingma, 2014) —— are taken from Song et al. (2021c) for both CIFAR-10, CelebA and LSUN/church_outdoor. For ImageNet2012, we reuse the parameter settings for CelebA.

# 6    CONCLUSION

This work demonstrated that introducing anisotropy into SBGM can be practically superior to traditional isotropic approaches. We extended SBGM by proposing a novel class of anisotropic diffusion processes theoretically founded on SPDEs. These processes generalize the conventional isotropic setting by enabling geometry-aware transformations that align the generative sampling dynamics more closely with intrinsic geometric structures in the data. Beyond the theoretical model, we presented a proof-of-concept implementation showing that this anisotropic framework can preserve fine-grained structural information over longer time scales and achieve competitive generative performance, supporting the underlying intuition. Exploring the broader parameter space of our anisotropic diffusion framework therefore constitutes a promising direction for future work. Together, these contributions broaden the design space of SBGMs and indicate that anisotropic transformations have the potential to further improve sample quality and convergence.

## 7 Reproducibility Statement

The theoretical framework underlying our approach is presented in Section 4. The specific instances used in our experiments, *Ours (isotropic)* and *Ours (anisotropic)*, are described in (17) and Section 5.2, including the corresponding parameter choices. The numerical simulation scheme for both forward and backward processes is detailed in Section H. Finally, the training hyperparameters and network architecture are specified in Section 5.4.

## 8 Ethics Statement

We acknowledge that diffusion-based generative models carry potential risks. In particular, they can be misused for generating deepfakes, which may facilitate misinformation, deception, or harassment, and they pose privacy risks if applied to generate images of individuals without consent. Our method is not intended for such purposes, and we ensured that our experiments relied solely on publicly available benchmark datasets without personally identifiable information. Moreover, our models are trained in a way that avoids replication or memorization of specific training images.

At the same time, diffusion models also offer positive contributions, including applications in medical imaging, scientific visualization, and artistic content creation. We believe that, with responsible use, the potential benefits of this line of research outweigh its risks.

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

# A   DESIGN GUIDELINE: CONTROLLING IMAGE TRANSFORMATIONS IN OUR FRAMEWORK

In this section, we provide design guidelines by detailing how the individual ingredients of our anisotropic diffusion framework control the image transformations described by our forward process.

## A.1   HOW DRIFT AND DIFFUSION COEFFICIENT CONTRIBUTE TO DEGRADING INFORMATION

The drift $b$ (7) introduces deterministic smoothing that respects the anisotropy defined by the *anisotropy coefficient* $\lambda_1$. This smoothing is controlled by the *diffusivity coefficient* $\alpha_1$, with its strength and direction determined by the local image structure (captured by the gradients $\nabla U_t$).

The diffusion coefficient $\sigma$ (8) injects random noise into the image, which is also modulated anisotropically by $\lambda_2$. The intensity of the noise is modulated by the *intensity coefficient* $\alpha_2$ and destroys fine-grained details in a controlled manner, complementing the drift's smoothing effect.

While the drift $b$ (7) focuses on smoothing (deterministic destruction), the diffusion coefficient $\sigma$ introduces stochasticity (random destruction). Together they degrade information in a controlled manner, providing an excellent framework for SBGM.

To enable (stochastic) reconstruction, we need to make sure that the forward process destroys the data sufficiently. Anisotropy can be seen as preserving information to some extent. This preserved data can be destroyed in various ways. For example, if the anisotropy is in the drift, we can either the destroy the information over time by spreading a lot of isotropic noise at later time points or by transitioning to isotropy in the drift. The latter can be achieved by letting $\lambda_1(t) \to \infty$ as $t \to T$.

Subsequently, we will give details on the effect of the diffusivity coefficients $\alpha_k$ (Section A.1.1), intensity coefficient (Section A.1.2), anisotropy coefficient $\lambda_i$ (Section A.1.3) and *correlation length* $\ell$ (Section E.2).

### A.1.1   DIFFUSIVITY COEFFICIENT $\alpha_1$

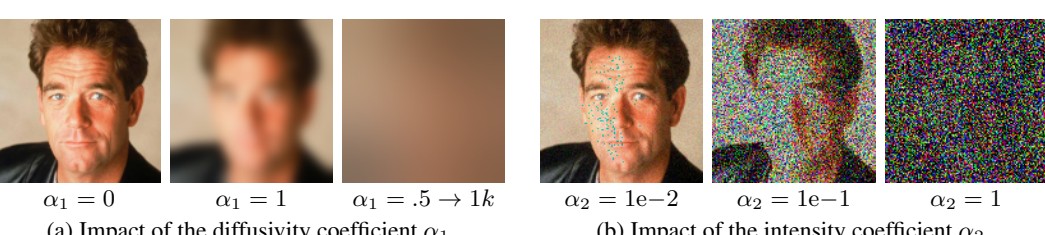

| $\alpha_1 = 0$ | $\alpha_1 = 1$ | $\alpha_1 = .5 \to 1k$ | $\alpha_2 = 1e{-}2$ | $\alpha_2 = 1e{-}1$ | $\alpha_2 = 1$ |

(a) Impact of the diffusivity coefficient $\alpha_1$.                    (b) Impact of the intensity coefficient $\alpha_2$.

The diffusivity coefficient $\alpha_1$ crucially controls the rate of the (an-)isotropic smoothing in the drift (7) in the diffusion coefficient (8) of the SPDE (6), respectively.

In the drift (7), a larger $\alpha_1$ results in stronger smoothing, leading to a faster elimination of high-frequency details (e.g., geometric structure, like edges and corners, and textures) in the image. Conversely, smaller values of $\alpha_1$ preserve more of the fine-grained details, slowing down the destruction of information.

In Figure (a) we visualized the impact of $\alpha_1$, disabling the effect of the diffusion coefficient $\sigma$ by setting $\alpha_2 = 0$. The third column is using a *geometric* transition $\alpha_1(t) = \alpha_1^{\min}\left(\alpha_1^{\max}/\alpha_1^{\min}\right)^{t/T}$ with $\alpha_1^{\min} = .5$ and $\alpha_1^{\max} = 1k$. In Figure 5 (b) we show more choices for the diffusivity/intensity coefficients. The original image had a resolution of 128x128 pixels and the forward process was simulated up to time $T = 8$.

### A.1.2   INTENSITY COEFFICIENT $\alpha_2$

The intensity coefficient $\alpha_2$ determines the intensity of the injected noise in the diffusion coefficient (8) of the SPDE (6).

In the diffusion coefficient (8), larger $\alpha_2$ increases the randomness in the image transformation, introducing more noise and accelerating the destruction of structured information. On the other hand, smaller $\alpha_2$ reduces the randomness, preserving some of the original structure while still degrading information.

In Figure 3b we visualized the impact of $\alpha_2$, disabling the effect of the drift $b$ by setting $\alpha_1 = 0$.

**Impact on generative modeling**   The progression of the $\alpha_i$ over time determines how quickly the image is degraded by the forward process.

By modulating $\alpha_i$, we can design a destruction process that aligns with the objective of generative modeling: creating a sequence of progressively smoother / less noise images while maintaining enough structure for the model to learn the reconstruction of meaningful samples with a high resemblance to the images from the dataset.

A suitable balance between $\alpha_1$ in the drift (7) and $\alpha_2$ in the diffusion diffusion coefficient (8) ensures that the degradation is smooth but irreversible, providing a structured data destruction trajectory for training the generative model.

### A.1.3 ANISOTROPY COEFFICIENT $\lambda_i$

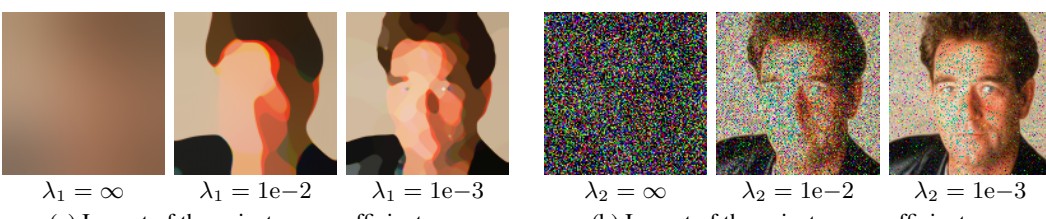

| $\lambda_1 = \infty$ | $\lambda_1 = 1e{-}2$ | $\lambda_1 = 1e{-}3$ | $\lambda_2 = \infty$ | $\lambda_2 = 1e{-}2$ | $\lambda_2 = 1e{-}3$ |

(a) Impact of the anisotropy coefficient $\alpha_1$.      (b) Impact of the anisotropy coefficient $\alpha_2$.

The anisotropy coefficients $\alpha_i$ control the directional sensitivity of both the drift and diffusion terms by modulating the degree of anisotropy in the transformation. The drift term (7) depends on $g_1(t, \nabla U_t)$, which introduces directional weighting to the smoothing property of the drift.

When $\lambda_1$ is small, the smoothing from the drift term (7) is highly anisotropic, meaning the process prefers certain directions for smoothing while preserving others (e.g., along edges in the image). This ensures that geometric structures are degraded in a structured manner. As $\lambda_1$ grows larger, the smoothing becomes more isotropic, uniformly degrading all directions and gradually eliminating all structural features.

The diffusion coefficient term (8), modulated by $g_2(t, \nabla U_t)$, introduces anisotropic noise. For smaller $\lambda_2$, the noise is injected along specific directions, preserving certain patterns while destroying others. As $\lambda_2$ increases, the noise becomes isotropic, introducing randomness uniformly across the image and further accelerating structured information loss. In the limit, when $\lambda_2 = \infty$, noise is injected isotropically.

In Figure 4a and Figure 4b we visualized the impact of $\lambda_1$ and $\lambda_2$, disabling the effect of the diffusion coefficient $\sigma$ and the drift $b$ by setting $\alpha_2 = 0$ and $\alpha_1 = 0$, respectively. In Figure 5 (a) we depicted common choices for these anisotropy coefficients.

**Impact on generative modeling**    The anisotropy coefficients $\lambda_i$ allow for a structured destruction of information. For example, by preserving geometric structures, like edges or corners, for longer time, the forward process provides richer intermediate representations, which can enhance the model's ability to reconstruct these structures during sampling.

Structured anisotropic degradation may lead to better generative sampling, as the score-based model learns to reverse transformations that align with natural image statistics (e.g., edge preservation and texture destruction).

In contrast, overly isotropic processes (corresponding to large up to infinite $\lambda_i$) degrade the images uniformly, which may simplify the forward process but could result in reduced resemblance of the dataset images, especially if they admit significant geometric patterns.

## B    INCORPORATING ANISOTROPY IN FLOW MATCHING (FM)

Flow matching methods (Lipman et al., 2022) offer a flexible framework for generative modeling by constructing probability paths between distributions without requiring explicit SPDE formulations. In particular, they support structured perturbations via non-isotropic noise, as demonstrated in edge-aware extensions such as (Vandersanden et al., 2024). However, these perturbations remain spatially uniform and are typically conditioned only on the initial dataset sample. This limits their ability to adapt to the evolving geometry of the sample during generation.

In contrast, our SPDE-based framework allows for spatially dependent, anisotropic diffusion that reacts dynamically to the evolving image gradients. As a result, structural features such as edges are preserved not only based on their presence in the initial image but also as they emerge, weaken, or shift throughout the transformation. For example, if an initially weak edge becomes stronger over time, our formulation naturally reduces diffusion across it. Conversely, if a previously strong edge fades, the diffusion increases, enabling appropriate smoothing. In flow matching, by contrast, structural information is fixed at the start of the transformation and cannot adapt to changes during generation. This may result in the preservation of features that should fade away or in the undesired blurring of structures that only become salient later in the process.

## C    CONDITIONAL GENERATIVE MODELING THROUGH ANISOTROPIC DIFFUSION BRIDGES

DDBM (Zhou et al., 2023) address a fundamentally different task from ours. Rather than generating images unconditionally from noise, DDBM learn mappings between two given image distributions by modeling the bridge dynamics connecting them. While their method can, in principle, be applied to unconditional generation by choosing noise as the source distribution, their framework is primarily designed for conditional tasks such as image-to-image translation or editing. In this first application of our framework, however, we focus exclusively on unconditional image generation and therefore do not include a direct comparison.

Extending our framework to enable bridging between arbitrary source and target distributions, akin to DDBMs is part of future work.

# D  SAMPLING FROM THE BACKWARD PROCESS

If $\left(\overline{U}_t\right)_{t\in\overline{I}}$ is a Markov process, sampling in the predictor step is done by following its transition dynamics. This is the case, for example, in DDPM and in the situation we encounter in our anisotropic diffusion framework from Section 4 as well.

In SDE-driven SBGMs, we are in the special situation, where the forward process is the solution to an SDE of the form

$$\mathrm{d}U_t = b(t, U_t)\,\mathrm{d}t + \sigma(t, U_t)\,\mathrm{d}W_t \quad \text{for all } t \in \overline{I} \tag{18}$$

for some $Q$-Wiener process. If a suitable regularity condition (Haussmann & Pardoux, 1986) is in place, then

$$\mathrm{d}\overline{U}_t = \overline{b}\big(t, \overline{U}_t\big)\,\mathrm{d}t + \overline{\sigma}\big(t, \overline{U}_t\big)\,\mathrm{d}W_t \quad \text{for all } t \in \overline{I}, \tag{19}$$

where

$$\overline{b}(t, u) := \operatorname{tr} \mathrm{D}_u \Sigma(T - t, u) + \Sigma(T - t, u)s(T - t, u) - b(T - t, u); \tag{20}$$
$$\overline{\sigma}(t, u) := \sigma(T - t, u) \tag{21}$$

for $(t, x) \in \overline{I} \times \mathbb{R}^D$ and

$$\Sigma := \left(\sigma Q^{\frac{1}{2}}\right)\left(\sigma Q^{\frac{1}{2}}\right)^*. \tag{22}$$

In this case, the sampling in the predictor step is performed using a method for the numerical solution of a SDE, with the Euler-Maruyama method (Kloeden & Platen, 1992) being the simplest approach.

**Discussion**  We emphasize that, in the practical application of our framework, it is the finite-dimensional SDE — obtained via the numerical scheme simulating our SPDE (6) described in Section H — that must be reversed in time, not the SPDE (6) itself. While — under a suitable set of assumptions — time-reversal of the SPDE (6) is theoretically possible (Föllmer & Wakolbinger, 1986; Millet et al., 1989), the results presented in Haussmann & Pardoux (1986); Anderson (1982) are sufficient for our purposes, as they apply directly to the finite-dimensional SDE setting.

# E  FURTHER VISUALIZATIONS OF THE
PARAMETERS OF OUR ANISOTROPIC DIFFUSION FRAMEWORK

## E.1  DIFFUSIVITY, INTENSITY AND ANISOTROPY COEFFICIENTS $\alpha_1$, $\alpha_2$ AND $\lambda_i$

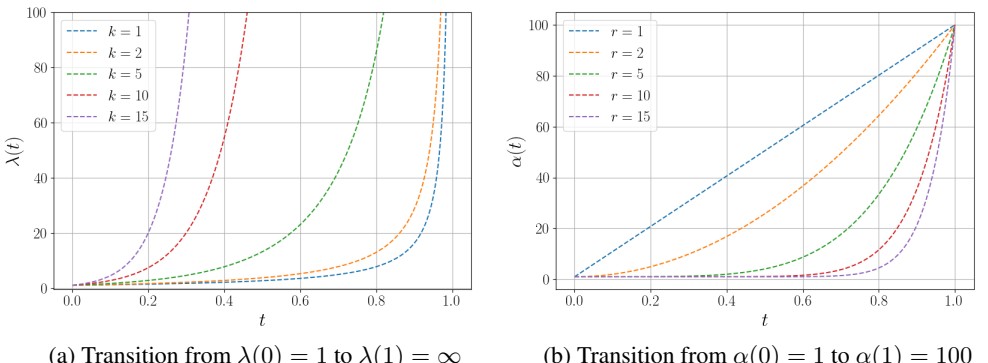

(a) Transition from $\lambda(0) = 1$ to $\lambda(1) = \infty$      (b) Transition from $\alpha(0) = 1$ to $\alpha(1) = 100$

Figure 5: (a) Visualization of the common choice $\lambda(t) := \lambda^{\min}\frac{e^{kT}-1}{e^{k(T-t)}-1}$ for the anisotropy coefficients (Section A.1.3). (b) Visualization of the common choice $\alpha(t) := \alpha^{\min} + \left(\alpha^{\max} - \alpha^{\min}\right)\left(\frac{t}{T}\right)^r$ for the diffusivity (Section A.1.1) and intensity coefficients (Section A.1.2).

### E.2 WIENER PROCESS COVARIANCE CORRELATION LENGTH $\ell$

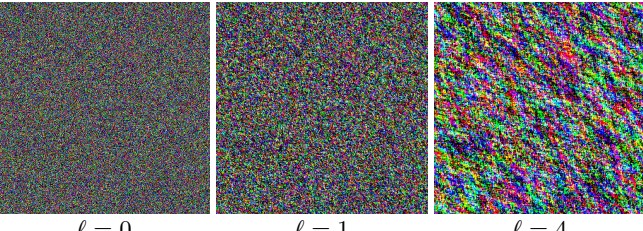

$$\ell = 0 \qquad\qquad \ell = 1 \qquad\qquad \ell = 4$$

Figure 6: Noise spread by the Wiener process $W$ for correlation lengths $\ell = 0, 1, 4$.

With the *correlation length* $\ell$ we effectively set the spatial extent of correlation among the spread noise. A large $\ell$ means noise at a given pixel is correlated with noise in a broader region around that pixel. In contrast, a small $\ell$ means the correlation is localized, and noise at a pixel primarily affects nearby pixels. A given pixel is roughly strongly (correlation $> .8$), moderately (correlation $> .5$) and weakly (correlation $> .1$ correlated to all pixels in a radius of $.22\ell$, $.69\ell$ and $2.3\ell$, respectively (cf. Figure 7). For our experiments in Section 5, we only considered the border case $\ell = 0$ in which (11) formally reduces to the Dirac delta $\delta_x(y)$ and hence $Q = \mathrm{id}_{L^2(\Lambda)}$ and hence $(W_t)_{t \in \overline{I}}$ is actually cylindrical. In Figure 6 we compared different choices for $\ell$.

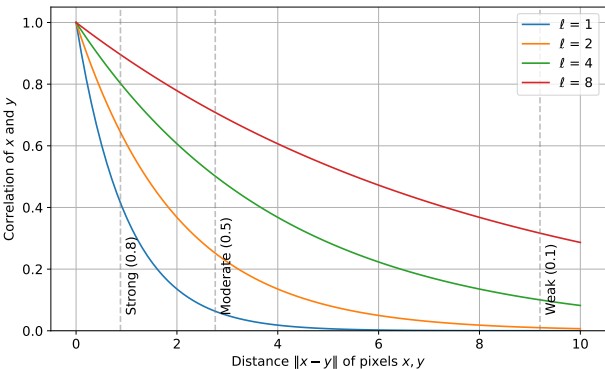

Figure 7: Spatial exponential decay of correlations in the Wiener process $W$ for correlation lengths $\ell = 1, 2, 4, 8$. For $\ell = 0$, the noise spread between pixels is independent.

## F  THE CAMERON-MARTIN SPACE $Q^{\frac{1}{2}} L^2(\Lambda)$

The space $Q^{\frac{1}{2}} L^2(\Lambda)$ is usually called a *Cameron-Martin space*; see (Da Prato & Zabczyk, 2014, Chapter I.4) or (Lord et al., 2014, Definition 10.15). To understand and apply our paper, it is only important to be aware of the set-theoretic definition $Q^{\frac{1}{2}} L^2(\Lambda) := \{Q^{\frac{1}{2}} u : u \in L^2(\Lambda)\}$. In plain English, it is the space of all transformations of $L^2(\Lambda)$-function under the operator $Q^{\frac{1}{2}}$. For more theoretical considerations, one important aspect is that it inherits a Hilbert space structure from $L^2(\Lambda)$.

## G  SPDE CLASSIFICATION

In general, (6) is a quasilinear parabolic SPDE with multiplicative noise.

If $g_1$ does not depend on the second argument, (6) is a semilinear parabolic SPDE with multiplicative noise:

$$\mathrm{d}U_t = \alpha_1(t)\Delta U_t \, \mathrm{d}t + \sigma(t, U_t) \, \mathrm{d}W_t \quad \text{for all } t \in \overline{I}. \tag{23}$$

If $g_2$ does not depend on the second argument, (6) is a quasilinear parabolic SPDE with additive noise:

$$\mathrm{d}U_t = b(t, U_t) \, \mathrm{d}t + \alpha_2(t) \, \mathrm{d}W_t \quad \text{for all } t \in \overline{I}. \tag{24}$$

Finally, if $g_1$ and $g_2$ both do not depend on the second argument, (6) is a linear parabolic SPDE with additive noise:

$$\mathrm{d}U_t = \alpha_1(t)\Delta U_t \, \mathrm{d}t + \alpha_2(t) \, \mathrm{d}W_t \quad \text{for all } t \in \overline{I}. \tag{25}$$

$$t = 0 \qquad t = \tfrac{1}{3}T \qquad t = \tfrac{2}{3}T \qquad t = T$$

Forward process (data destruction)

Backward process (data generation)

Figure 8: Visualization of the forward and backward processes corresponding to our anisotropic stochastic heat equation with isotropic noise (Section 5.2).

## H    NUMERICAL SIMULATION

For the numerical simulation of the forward and backward processes, (6) and (1), we modeled the image space $\Lambda$ as $\Lambda = (0, d_1) \times (0, d_2)$ and decomposed the boundary $\partial\Lambda$ according to

$$\partial_L \Lambda := \{0\} \times [0, d_2); \tag{26}$$
$$\partial_T \Lambda := [0, d_1) \times \{d_2\}; \tag{27}$$
$$\partial_R \Lambda := \{d_1\} \times (0, d_2]; \tag{28}$$
$$\partial_B \Lambda := (0, d_1] \times \{0\} \tag{29}$$

into its left, top, right and bottom part. We discretized the derivatives using a mixture of forward, backward and central finite differences, respecting Neumann boundary conditions.

### H.1    DOMAIN DISCRETIZATION

After discretization, we decomposed the discretized domain $D = \{0, \ldots, d_1\} \times \{0, \ldots, d_2\}$ in the same spirit into its interior, left, top, right and bottom part:

$$D^\circ := \{1, \ldots, d_1 - 2\} \times \{1, \ldots, d_2 - 2\}; \tag{30}$$
$$\partial_L D := \{0\} \times \{0, \ldots, d_2 - 2\}); \tag{31}$$
$$\partial_T D := \{0, \ldots, d_2 - 2\} \times \{d_2 - 1\}; \tag{32}$$
$$\partial_R D := \{d_1 - 1\} \times \{1, \ldots, d_2 - 1\}; \tag{33}$$
$$\partial_B D := \{1, \ldots, d_1 - 1\} \times \{0\}. \tag{34}$$

## H.2 SPATIAL DISCRETIZATION

For the specific finite difference approximation we have chosen, we ended up with the discretized drift being given by

$$
\tilde{b}(t,u)_i := \begin{cases}
\begin{aligned}
& g_1\left(t, \begin{pmatrix} u_{i_1+2 \wedge d_1-1,\, i_2} - u_i \\ u_{i_1+1,\, i_2+1} - u_{i_1+1,\, i_2-1} \end{pmatrix}\right)(u_{i_1+1,\, i_2} - u_i) \\
& - g_1\left(t, \begin{pmatrix} u_i - u_{i_1-2 \vee 0,\, i_2} \\ u_{i_1-1,\, i_2+1} - u_{i_1-1,\, i_2-1} \end{pmatrix}\right)(u_i - u_{i_1-1,\, i_2}) \\
& + g_1\left(t, \begin{pmatrix} u_{i_1+1,\, i_2+1} - u_{i_1-1,\, i_2+1} \\ u_{i_1,\, i_2+2 \wedge d_2-1} - u_i \end{pmatrix}\right)(u_{i_1,\, i_2+1} - u_i) \\
& - g_1\left(t, \begin{pmatrix} u_{i_1+1,\, i_2-1} - u_{i_1-1,\, i_2-1} \\ u_i - u_{i_1,\, i_2-2 \vee 0} \end{pmatrix}\right)(u_i - u_{i_1,\, i_2-1})
\end{aligned} & , \text{if } i \in D^\circ; \\[2em]
\begin{aligned}
& \left(g_1\left(t, \begin{pmatrix} u_{i_1+2 \wedge d_1-1,\, i_2} - u_i \\ 0 \end{pmatrix}\right) + g_1(t,0)\right)(u_{i_1+1,\, i_2} - u_i) \\
& + \left(g_1\left(t\begin{pmatrix} 0 \\ u_{i_1,\, i_2+2 \wedge d_2-1} - u_i \end{pmatrix}\right) + g_1(t,0)\right)(u_{i_1,\, i_2+1} - u_i)
\end{aligned} & , \text{if } i \in \partial_L D \text{ with } i_2 = 0; \\[2em]
\begin{aligned}
& \left(g_1\left(t, \begin{pmatrix} u_{i_1+2 \wedge d_1-1,\, i_2} - u_i \\ u_{i_1+1,\, i_2+1} - u_{i_1+1,\, i_2-1} \end{pmatrix}\right)\right. \\
& \left. + g_1\left(t, \begin{pmatrix} 0 \\ u_{i_1+1,\, i_2+1} - u_{i_1+1,\, i_2-1} \end{pmatrix}\right)\right)(u_{i_1+1,\, i_2} - u_i) \\
& + g_1\left(t, \begin{pmatrix} 0 \\ u_{i_1,\, i_2+2 \wedge D_2-1} - u_i \end{pmatrix}\right)(u_{i_1,\, i_2+1} - u_i) \\
& - g_1\left(t, \begin{pmatrix} 0 \\ u_i - u_{i_1,\, i_2-2 \vee 0} \end{pmatrix}\right)(u_i - u_{1,\, 2-1})
\end{aligned} & , \text{if } i \in \partial_L D \text{ with } i_2 > 0; \\[2em]
\begin{aligned}
& \left(g_1\left(t, \begin{pmatrix} u_{i_1+2 \wedge d_1-1,\, i_2} - u_i \\ 0 \end{pmatrix}\right) + g_1(t,0)\right)(u_{i_1+1,\, i_2} - u_i) \\
& + \left(g_1\left(t, \begin{pmatrix} 0 \\ u_i - u_{i_1,\, i_2 \vee 0} \end{pmatrix}\right) + g_1(t,0)\right)(u_{i_1,\, i_2-1} - u_i)
\end{aligned} & , \text{if } i \in \partial_T D \text{ with } i_1 = 0; \\[2em]
\begin{aligned}
& g_1\left(t, \begin{pmatrix} u_{i_1+2 \wedge d_1-1,\, i_2} - u_i \\ 0 \end{pmatrix}\right) \\
& - g_1\left(t, \begin{pmatrix} u_i - u_{i_1-2 \vee 0,\, i_2} \\ 0 \end{pmatrix}\right)(u_i - u_{i_1-1,\, i_2}) \\
& + \left(g_1\left(t, \begin{pmatrix} u_{i_1+1,\, i_2-1} - u_{i_1-1,\, i_2-1} \\ 0 \end{pmatrix}\right)\right. \\
& \left. + g_1\left(\begin{pmatrix} u_{i_1+1,\, i_2-1} - u_{i_1-1,\, i_2-1} \\ u_i - u_{i_1,\, i_2-2 \vee 0} \end{pmatrix}\right)\right)(u_{i_1,\, i_2-1} - u_i)
\end{aligned} & , \text{if } i \in \partial_T D \text{ with } i_1 > 0
\end{cases}
$$

$$\tag{35}$$

and

$$\tilde{b}(t,u)_i := \begin{cases} \left(g_1\left(t, \begin{pmatrix} u_i - u_{i_1-2 \vee 0,\, i_2} \\ 0 \end{pmatrix}\right) + g_1(t,0)\right)(u_{i_1-1,\, i_2+1} - u_i) \\ \qquad + \left(g_1\left(t, \begin{pmatrix} 0 \\ u_i - u_{i_1,\, i_2-2 \vee 0} \end{pmatrix}\right) + g_1(t,0)\right)(u_{i_1,\, i_2-1} - u_i) & , \text{if } i \in \partial_R D \text{ with } i_2 = d_2 - 1; \\[2mm] \left(g_1\left(t, \begin{pmatrix} 0 \\ u_{i_1-1,\, i_2+1} - u_{i_1-1,\, i_2-1} \end{pmatrix}\right) \right. \\ \qquad \left. + g_1\left(t, \begin{pmatrix} u_i - u_{i_1-2 \vee 0,\, i_2} \\ u_{i_1-1,\, i_2+1} - u_{i_1-1,\, i_2-1} \end{pmatrix}\right)\right)(u_{i_1-1,\, i_2} - u_i) \\ \qquad + g_1\left(t, \begin{pmatrix} 0 \\ u_{i_1,\, i_2+2 \wedge d_2-1} - u_i \end{pmatrix}\right)(u_{i_1,\, i_2+1} - u_i) \\ \qquad - g_1\left(t, \begin{pmatrix} 0 \\ u_i - u_{i_1,\, i_2-2 \vee 0} \end{pmatrix}\right)(u_i - u_{i_1,\, i_2-1}) & , \text{if } i \in \partial_R D \text{ with } i_2 < d_2 - 1; \\[2mm] \left(g_1\left(t, \begin{pmatrix} u_i - u_{i_1-2 \vee 0,\, i_2} \\ 0 \end{pmatrix}\right) + g_1(t,0)\right)(u_{i_1-1,\, i_2} - u_i) \\ \qquad + \left(g_1\left(t, \begin{pmatrix} 0 \\ u_{i_1,\, i_2+2 \wedge d_2-1} - u_i \end{pmatrix}\right) + g_1(t,0)\right)(u_{i_1,\, i_2+1} - u_i) & , \text{if } i \in \partial_B D \text{ with } i_1 = d_1 - 1; \\[2mm] g_1\left(t, \begin{pmatrix} u_{i_1+2 \wedge d_1-1,\, i_2} - u_i \\ 0 \end{pmatrix}\right)(u_{i_1+1,\, i_2} - u_i) \\ \qquad - g_1\left(t, \begin{pmatrix} u_i - u_{i_1-2 \vee 0,\, i_2} \\ 0 \end{pmatrix}\right)(u_i - u_{i_1-1,\, i_2}) \\ \qquad + \left(g_1\left(t, \begin{pmatrix} u_{i_1+1,\, i_2+1} - u_{i_1-1,\, i_2+1} \\ u_{i_1,\, i_2+2 \wedge d_2-1} - u_i \end{pmatrix}\right)\right. \\ \qquad \left. + g_1\left(t, \begin{pmatrix} u_{i_1+1,\, i_2+1} - u_{i_1-1,\, i_2+1} \\ 0 \end{pmatrix}\right)\right)(u_{i_1,\, i_2+1} - u_i) & , \text{if } i \in \partial_B D \text{ with } i_1 < d_1 - 1 \end{cases}$$

(36)

for $(t,u) \in \overline{I} \times \mathbb{R}^D$ and $i \in D$ and the discretized diffusion coefficient being given by

$$(\tilde{\sigma}(t,u)v)_i := \begin{cases} g_2\left(t, \begin{pmatrix} u_{i_1+1,\, i_2} - u_{i_1-1, i_2} \\ u_{i_1, i_2+1} - u_{i_1, i_2-1} \end{pmatrix}\right) & , \text{if } i \in D^\circ; \\[2mm] g_2\left(t, \begin{pmatrix} 0 \\ u_{i_1, i_2+1} - u_{i_1, i_2-1} \end{pmatrix}\right) & , \text{if } i \in \partial_L D \text{ with } i_2 > 0 \text{ or} \\ & \quad\ i \in \partial_R D \text{ with } i_2 < d_2 - 1; \\[2mm] g_2\left(t, \begin{pmatrix} u_{i_1+1, i_2} - u_{i_1-1, i_2} \\ 0 \end{pmatrix}\right) & , \text{if } i \in \partial_T D \text{ with } i_1 > 0 \text{ or} \\ & \quad\ i \in \partial_B D \text{ with } i_1 < d_1 - 1; \\[2mm] g_2(t,0) & , \text{if } i \in \partial_L D \text{ with } i_2 = 0 \text{ or} \\ & \quad\ i \in \partial_T D \text{ with } i_1 = 0 \text{ or} \\ & \quad\ i \in \partial_R D \text{ with } i_2 = d_2 - 1 \text{ or} \\ & \quad\ i \in \partial_B D \text{ with } i_1 = d_1 - 1 \end{cases} v_i \quad (37)$$

for $v \in \mathbb{R}^D$, $(t,u) \in \overline{I} \times \mathbb{R}^D$ and $i \in D$.

## H.3 TEMPORAL DISCRETIZATION

For temporal discretization, we used a *drift-termed* (explicit) Euler-Maruyama scheme (Hutzenthaler & Jentzen, 2015), where given a generic SDE of the form

$$d\tilde{U}_t = \tilde{b}\left(t, \tilde{U}_t\right) dt + \tilde{\sigma}\left(t, \tilde{U}_t\right) d\tilde{W}_t \quad \text{for all } t \in \overline{I}, \tag{38}$$

the time stepping is given by

$$\tilde{U}_{t+\Delta t} = \tilde{U}_t + \frac{b\left(t, \tilde{U}_t\right)}{1 + \Delta t \left\| b\left(t, \tilde{U}_t\right)\right\|^{\gamma}} + \tilde{\sigma}\left(t, \tilde{U}_t\right)\left(\tilde{W}_{t+\Delta t} - \tilde{W}_t\right) \tag{39}$$

for all $t, \Delta t \geq 0$ with $t + \Delta t \in \overline{I}$, where $\gamma$ is a *taming coefficient* usually chosen to be 1.

# I SPDE TRAJECTORY VISUALIZATIONS

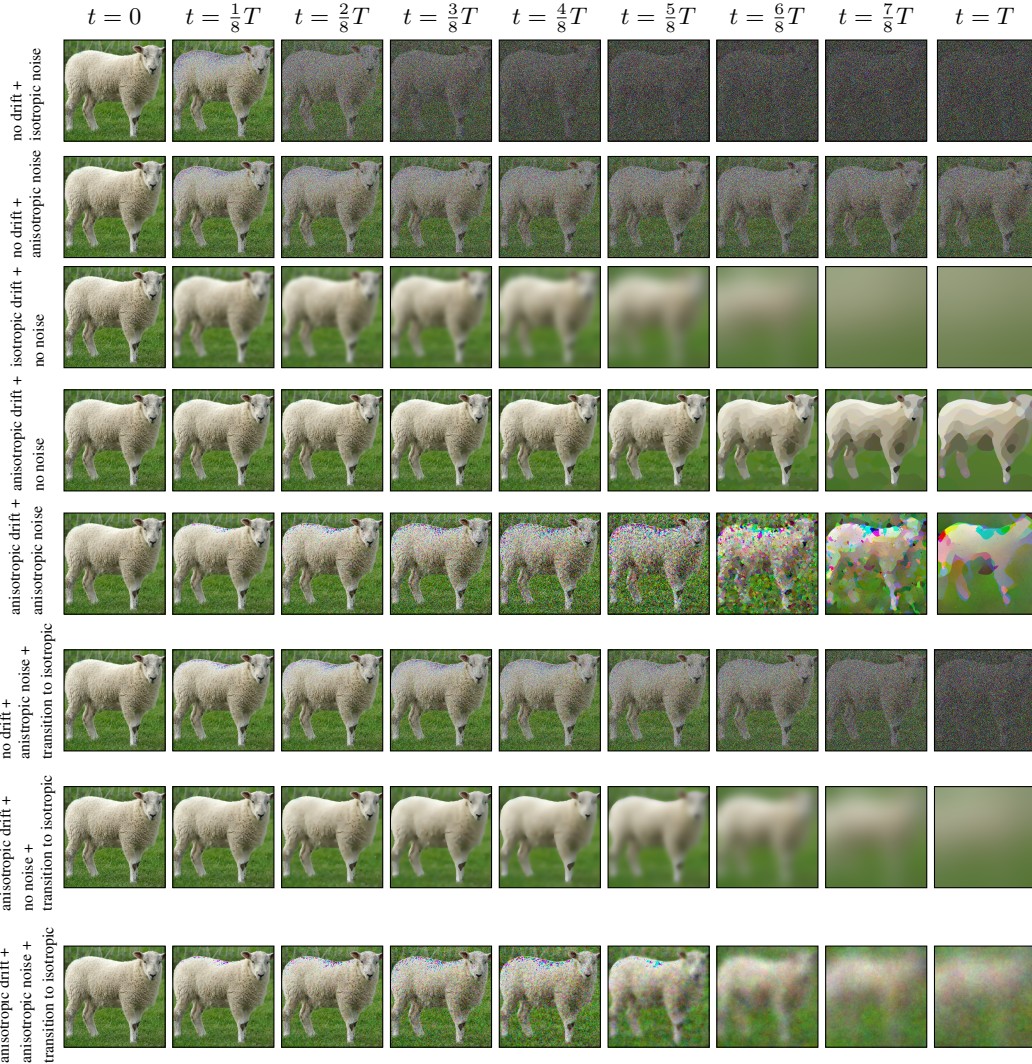

Figure 9: We visualize the core ingredients of our generalized anisotropic SPDE diffusion framework. The diffusion process is governed by two fundamental components: the drift term, driven by the drift coefficient $b$ and the diffusion term, driven by the diffusion coefficient $\sigma$. Both terms can take on isotropic or anisotropic forms, and their combinations open the door to a vast spectrum of processes. These processes destroy (and regenerate, for the reverse process) the signal's information in ways that range from subtle to profoundly distinct. The interplay between these terms offers the designer of the generative process fine control over how information is destroyed.

## J  UNCURATED GENERATED SAMPLES ON CIFAR-10

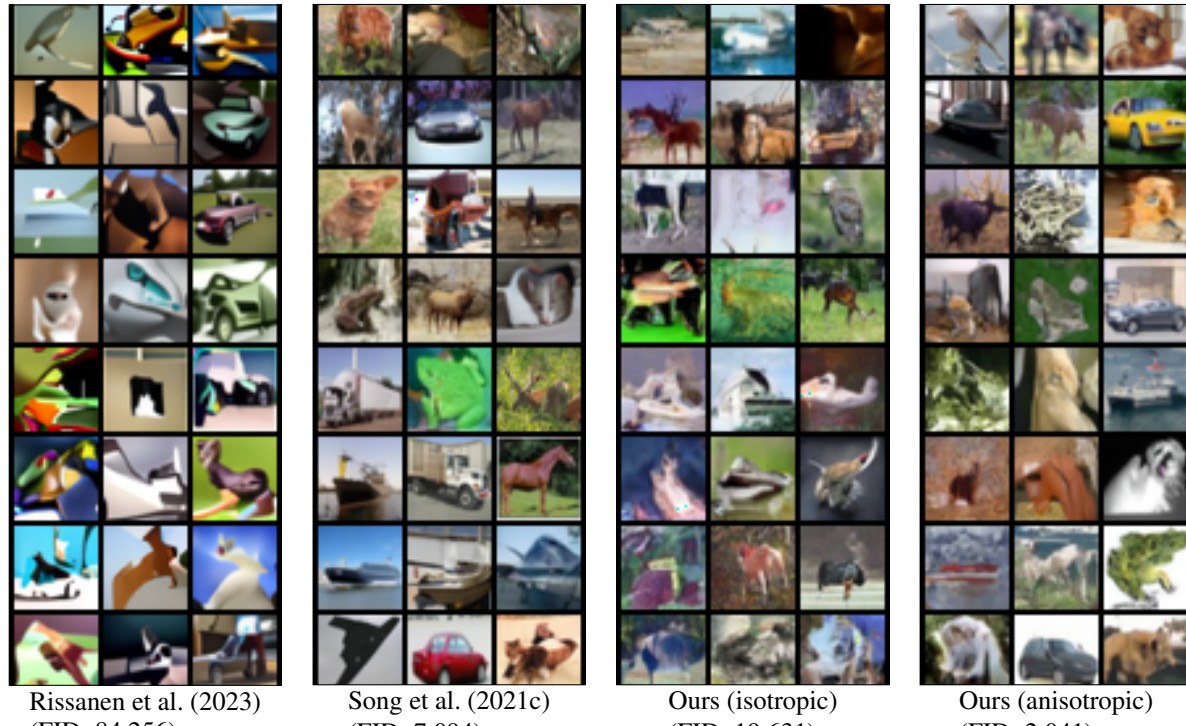

| Rissanen et al. (2023) | Song et al. (2021c) | Ours (isotropic) | Ours (anisotropic) |
| (FID: 84.256) | (FID: 7.094) | (FID: 19.631) | (FID: 2.041) |

Figure 10: Uncurated samples for the baselines Rissanen et al. (2023), Song et al. (2021c) and two of our SPDEs: one with isotropic drift $b$ and isotropic diffusion coefficient $\sigma$ (see Section 5.3) and one with anisotropic $b$ and isotropic $\sigma$ (see Section 5.2). Note that Rissanen et al. (2023) and *Ours (isotropic)* essentially represent the same SPDE, but our score-based approach performs significantly better.

## K    UNCURATED GENERATED SAMPLES ON CELEBA

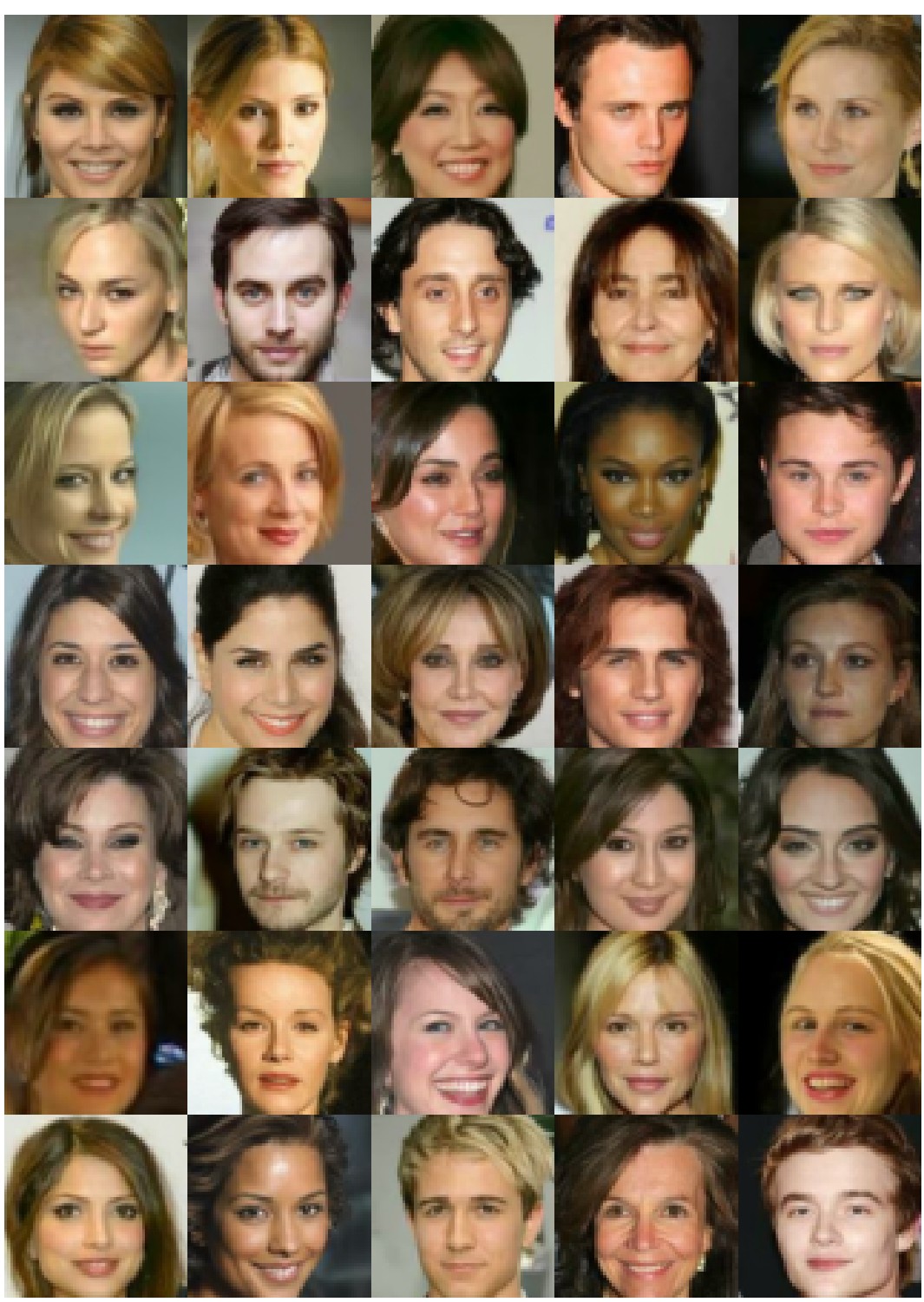

Figure 11: Uncurated samples for *Ours (anisotropic)* (see Section 5.2). The generated images are produced by a model trained from scratch — without initialization from a pre-trained network — for 100,000 iterations.

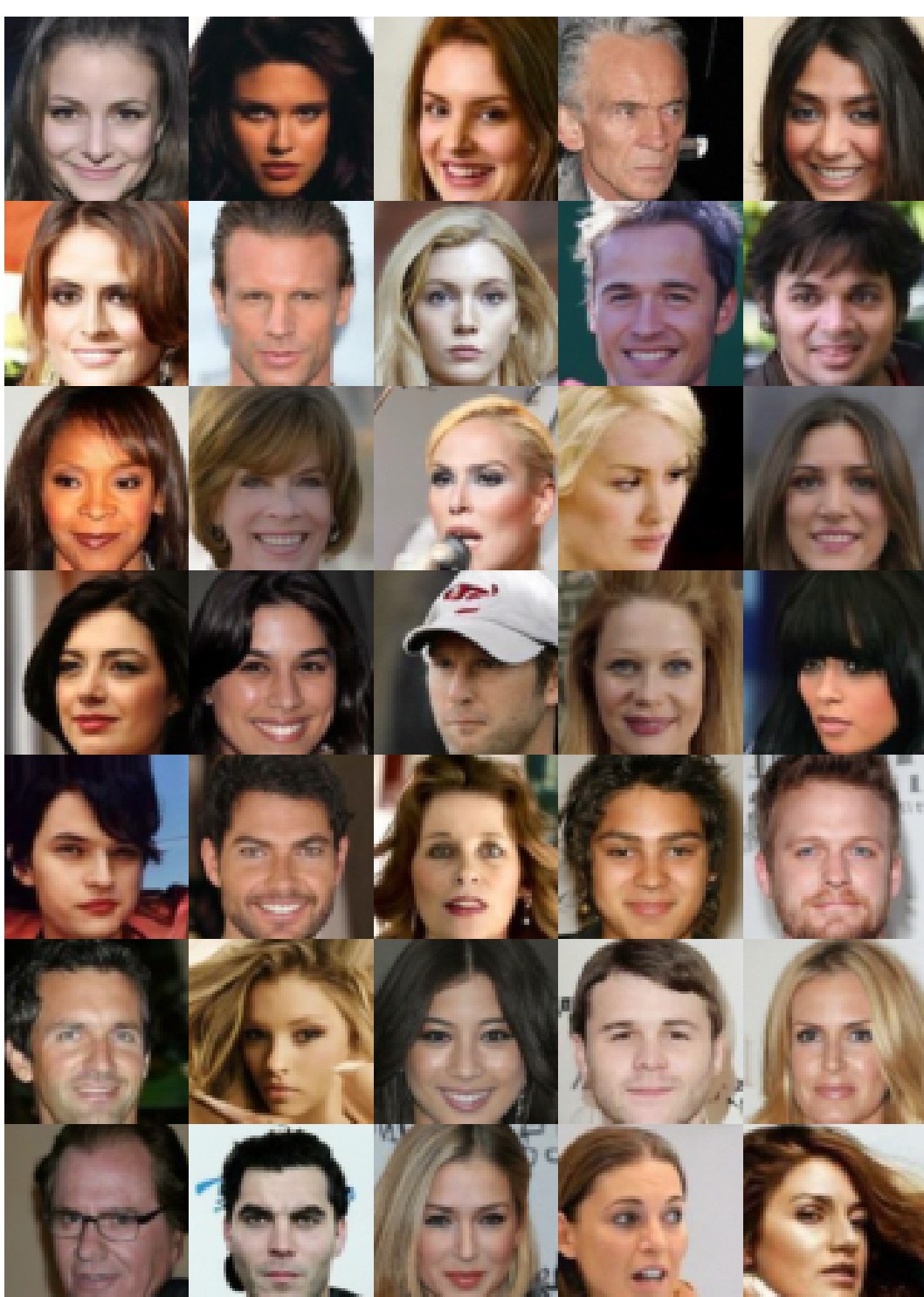

Figure 12: Uncurated samples for Song et al. (2021c). The generated images are produced from the checkpoint provided by the authors.

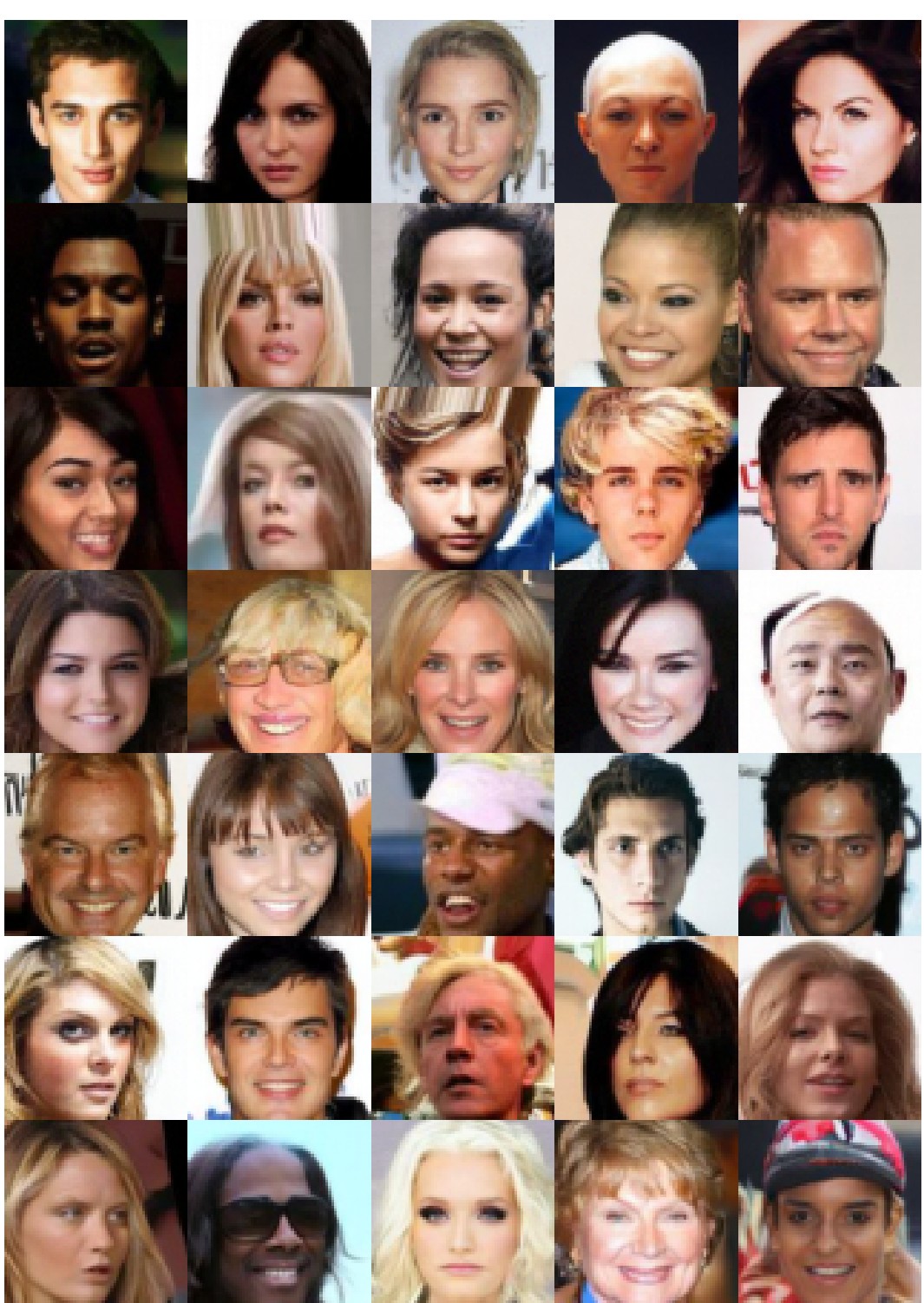

Figure 13: Uncurated samples for Lipman et al. (2022). The generated images are produced by a model trained from scratch — without initialization from a pre-trained network — for 100,000 iterations.

## L UNCURATED GENERATED SAMPLES ON IMAGENET2012

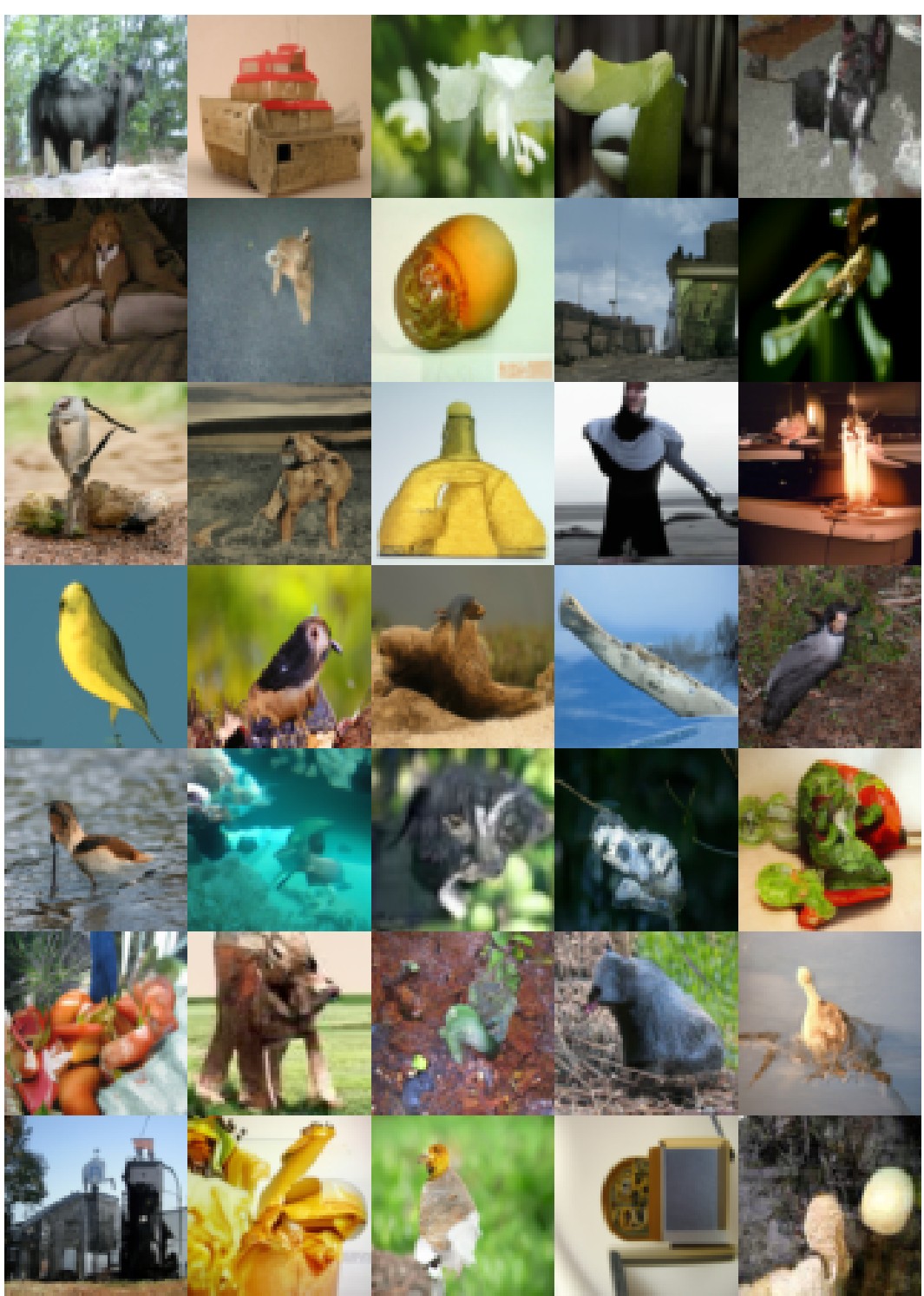

Figure 14: Uncurated samples for *Ours (anisotropic)* (see Section 5.2). The generated images are produced by a model trained from scratch — without initialization from a pre-trained network — for 100,000 iterations.

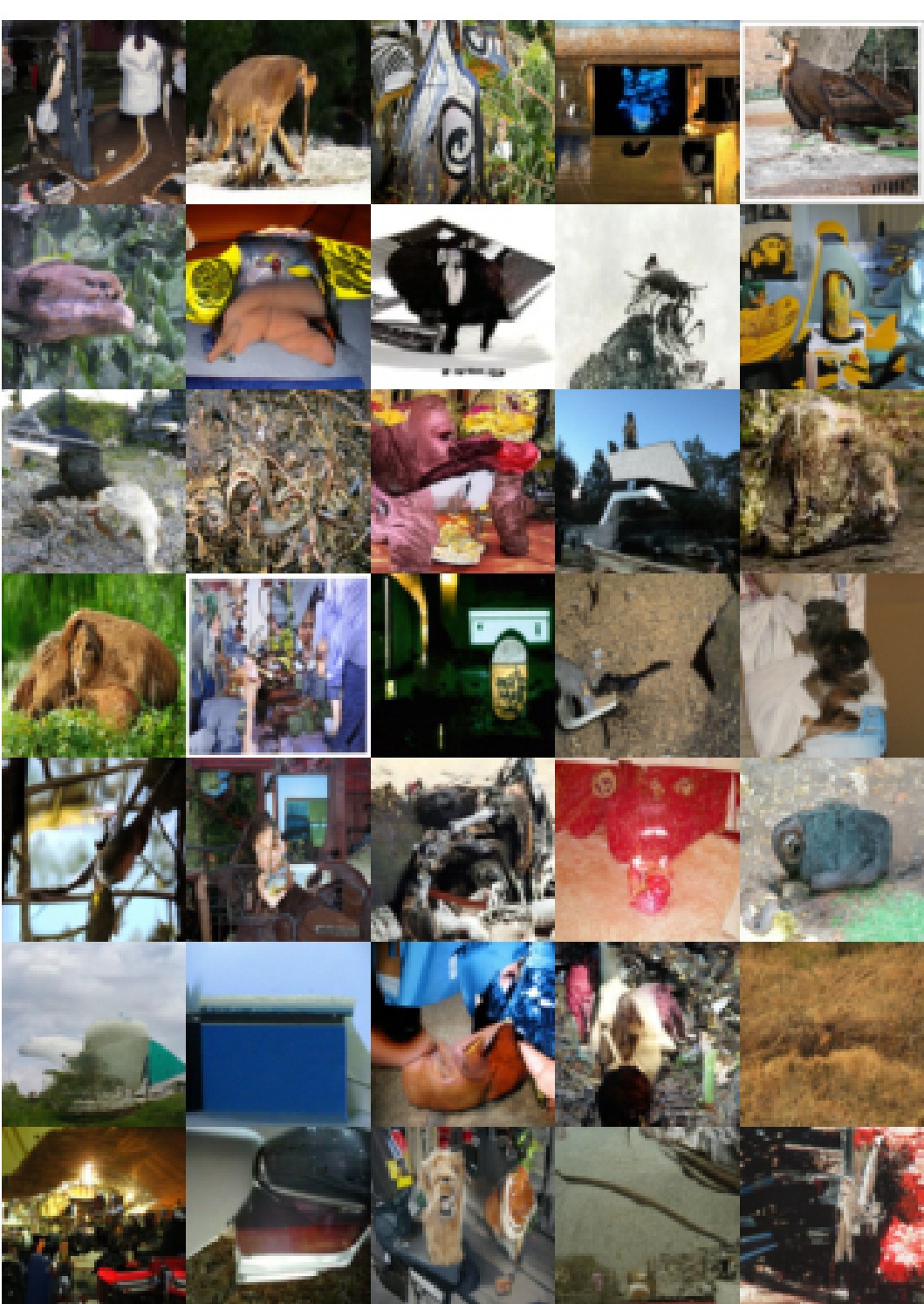

Figure 15: Uncurated samples for Song et al. (2021c). The generated images are produced from the checkpoint provided by the authors.

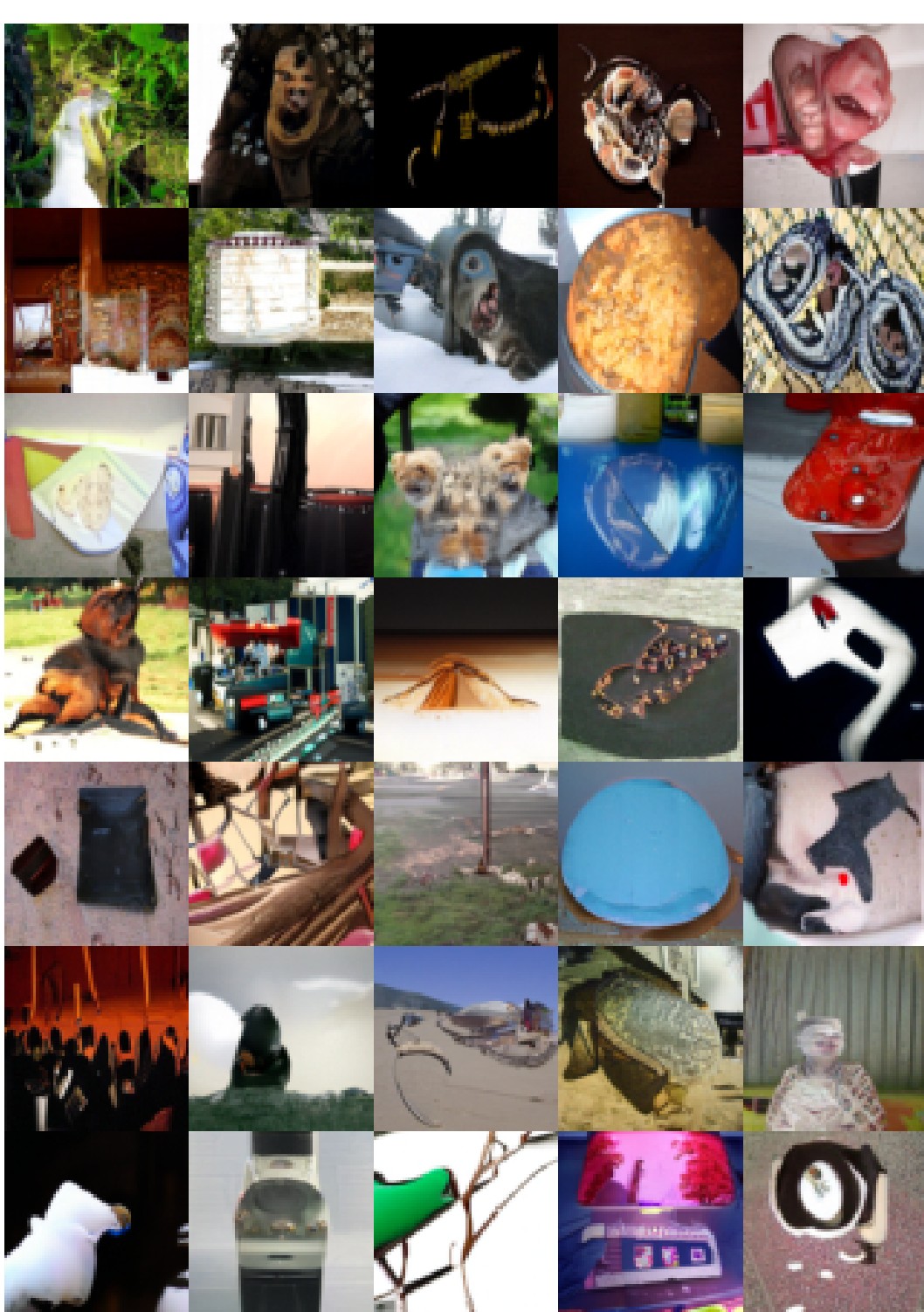

Figure 16: Uncurated samples for Lipman et al. (2022). The generated images are produced by a model trained from scratch — without initialization from a pre-trained network — for 100,000 iterations.

## M    UNCURATED GENERATED SAMPLES ON LSUN/CHURCH_OUTDOOR

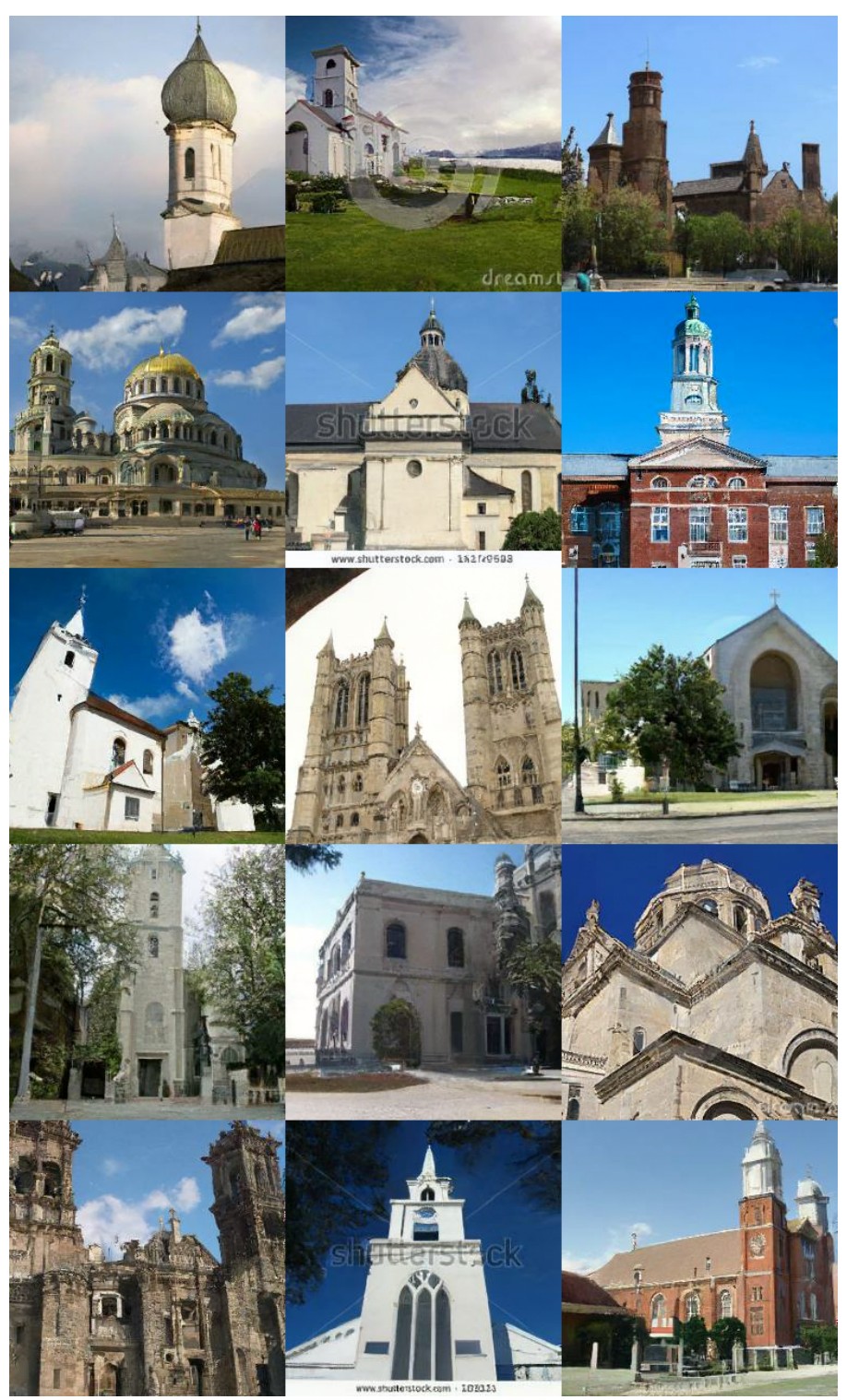

Figure 17: Uncurated samples for *Ours (anisotropic)* (see Section 5.2). The generated images are produced by a model trained from scratch — without initialization from a pre-trained network — for 100,000 iterations.

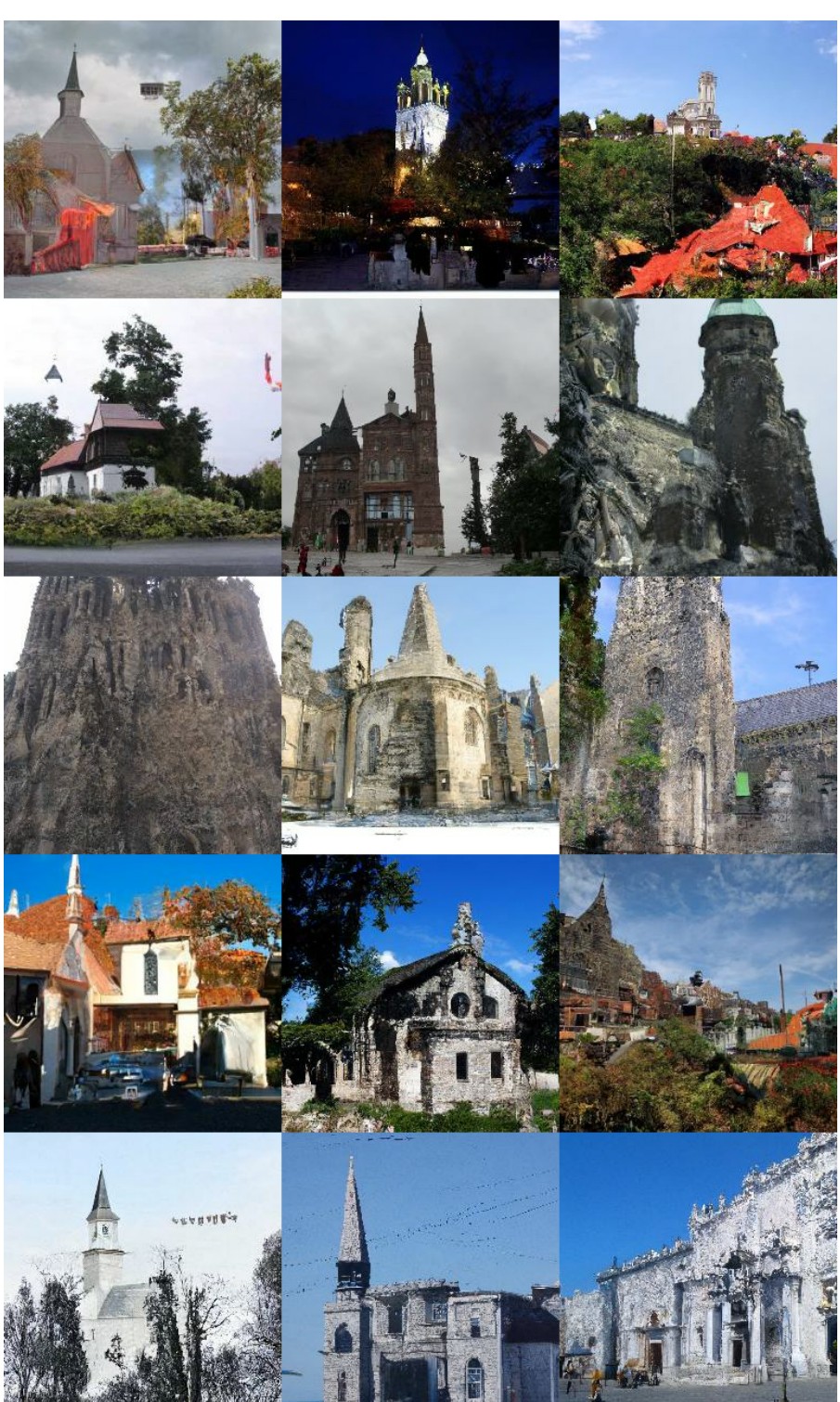

Figure 18: Uncurated samples for Song et al. (2021c). The generated images are produced from the checkpoint provided by the authors.

# N    COMPUTATIONAL COSTS

To contextualize our computational costs relative to Song et al. (2021c), we report the normalized training times per 10k steps on CelebA (64×64):

| Model | Time / 10k steps (s) |
|-------|---------------------|
| Song et al. (2021c) | 2506.55 |
| Ours (anisotropic) | 4658.63 |

Additionally, the following inference costs arise during sample generation:

| Model | Time / generated image (s) |
|-------|----------------------------|
| Song et al. (2021c) | 0.5726 |
| Ours (anisotropic) | 0.1649 |

As these timings indicate, our method achieves significantly improved inference times compared to Song et al. (2021c), due to our numerical implementation explained in Appendix H —— despite having a theoretically more demanding drift and diffusion coefficient.

# O    LATENT SPACE EXPERIMENT

We conducted a latent space experiment using *Ours (anisotropic)* on the LSUN/church_outdoor dataset. For the latent representation, we employed the pretrained variational autoencoder from `stabilityai/sd-vae-ft-mse`. The resulting generative performance metrics are:

$$\text{Inception Score} = 3.880973$$
$$\text{FID} = 3.936322$$
$$\text{KID} = 2.387085 \times 10^{-3}$$

These results demonstrate that our anisotropic SPDE framework can be successfully applied in latent space as well.

## P UNCURATED GENERATED SAMPLES ON LSUN/CHURCH_OUTDOOR FROM OUR LATENT SPACE EXPERIMENT

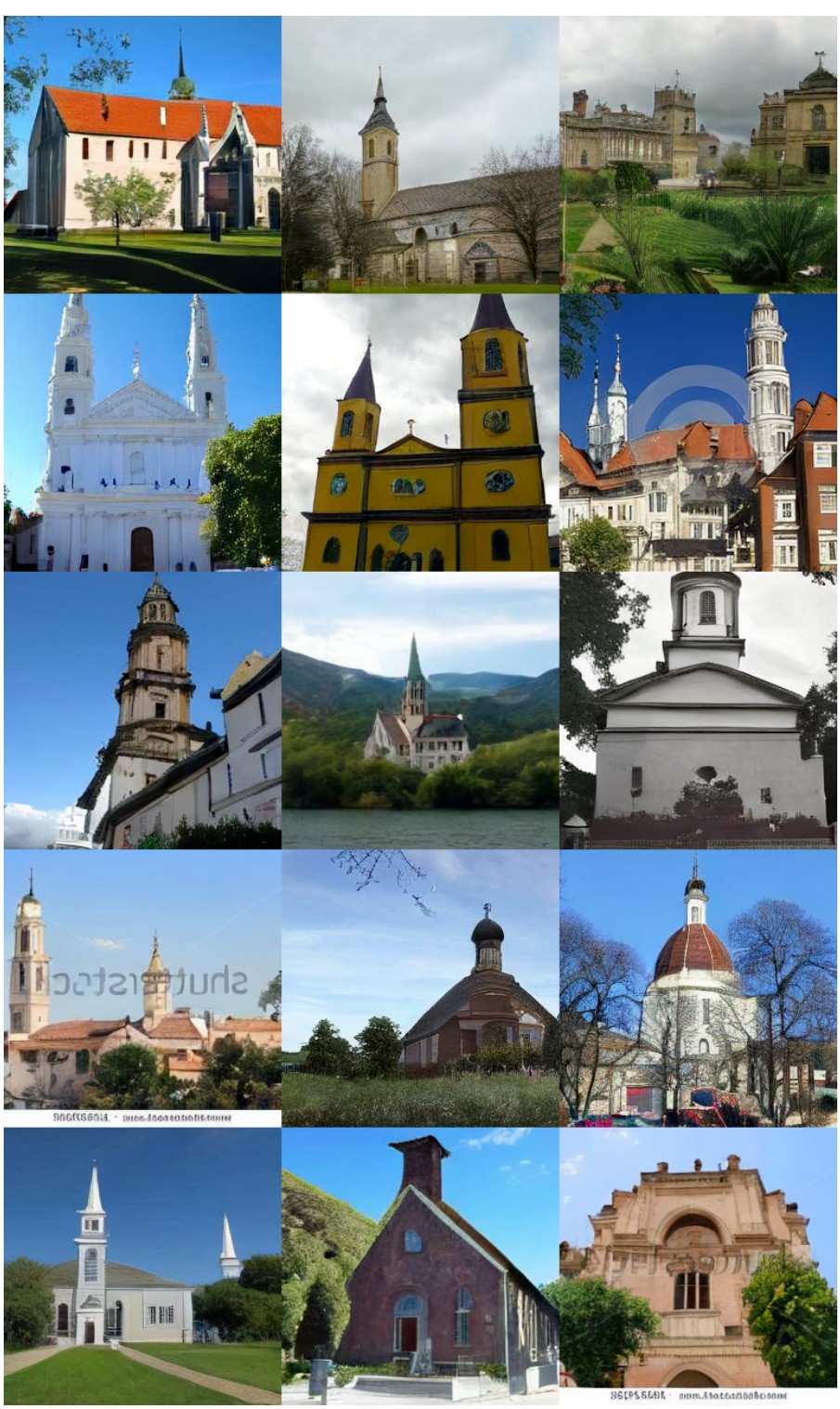

Figure 19: Uncurated samples for *Ours (anisotropic)* (see Section 5.2) from our latent space experiment. The generated images are produced by a model trained from scratch — without initialization from a pre-trained network — for 100,000 iterations.

