# OpenReview forum: "Score-based generative modeling through anisotropic SPDEs"
_ICLR.cc/2026/Conference — ICLR 2026 Conference Withdrawn Submission_

### Official Review · Reviewer_PX5s · 2025-10-28

**Soundness:** 2
**Presentation:** 1
**Contribution:** 1
**Rating:** 2
**Confidence:** 5

**Summary:**

This paper proposes an anisotropic nonlinear diffusion process. The authors claim to allow both the drift and diffusion coefficients to evolve dynamically based on the current state. However, their implementation in this paper appears to focus solely on the drift term. While stochastic differential equations (SDEs) with state-dependent drift and diffusion coefficients are well-established in theory, in my view, incorporating such dynamics into a generative diffusion model requires a rigorous framework to demonstrate that the backward diffusion process can recover the initial prior distribution. However, I could not find a clear explanation  of such framework presented in this paper. It needs a major revision.

**Strengths:**

This paper proposes an anisotropic nonlinear diffusion process.

**Weaknesses:**

This paper is poorly written. It does not flow well. It lacks clarity and coherence, making it difficult to follow.  I listed below some examples of specific problems:
1. The authors used non-standard  statistical terms such as "$\mu$-distributed sequence"
2. P3. Line 128, That authors stated that, during the forward process, "the dynamics
of the transformation realized by the forward process are learned (by a neural network)". This seems like a wrong statement.
3. Lack proper reference:  anisotropic nonlinear diffusion process has been investigate intensively in multiscale image analysis, however, this paper did not discuss the relevant of this topic in their introduction. Here are some references:
 1).  P Perona and J Malik, Scale-space and edge detection using anisotropic diffusion.
 2).  P. Guidotti  Anisotropic Diffusions of Image Processing From Perona-Malik
 3). Y Bao; H. Krim Smart nonlinear diffusion: a probabilistic approach
:4). W Feng, P Qiao, X Xi, and Y Chen,  Image Denoising via Multiscale Nonlinear Diffusion Models
 5). Y You , W Xu, A Tannenbaum, M Kaveh, Behavioral analysis of anisotropic diffusion in image processing

4. Repeating:
for example P7, Line 361: We now begin by describing specific instances of our framework
                    P7, Line 363: We now describe the specific instances of our framework considered
5, The selection of drift and diffusion coefficients in Eqns (7)-(9): The definition of sigma in eqn. (8) was not used in the new algorithm in section 5.2 and 5.3. More importantly, there is no proof on why the selection of the coefficients will lead to preserving structures in images.

**Questions:**

1. Is there a timeline problem in the following statement:
P2. Lin 90: The authors stated that:
Rissanen et al. (2023) considered a stochastic heat equation with isotropic noise, which is effectively destroying the data by blurring up to complete dissipation. This is in contrast to earlier approaches that typically destroyeddata into pure noise. Hoogeboom & Salimans (2022) extended this idea by introducing a temporally increasing isotropic noise term, further refining the blurring process over time.

2. Eqns. (4) (5) is not used anywhere, any discussion on this and how it is used in the proposed algorithm?

3.There is a $v$ in Eqn. (8), but there is no $v$ given in Eqn. (6).

---

> ### Author Response · Authors · 2025-11-17
>
> We thank the reviewer for the careful examination of our work and for the valuable suggestions for improvement.
>
> # Does our implementation focus solely on the drift term, and why is it guaranteed that the backward process is distributed according to the data distribution at the terminal point?
>
> Our proposed SPDE (Eq. 6) contains both a time- and space-gradient-dependent drift and a diffusion coefficient. As explained in detail in Section 4.2, numerically simulating an SPDE of this generality necessarily requires simulating a finite-dimensional SDE that approximates it; in our case this is Eq. 12 (or Eq. 38 in the appendix). For such finite-dimensional SDEs, the well-known results on SDE time-reversal apply, showing that the time-reversed process again satisfies an SDE. In our case, this corresponds exactly to Eq. 19.
>
> ---
>
> # What is a “$\mu$-distributed sequence”?
>
> This is standard terminology in probability theory. It refers to a sequence of identically distributed random variables whose common distribution is $\mu$.
>
> ---
>
> # Is it appropriate to claim that during the forward process, the dynamics of the transformation realized by the forward process are learned?
>
> We agree that the sentence in Line 128 is misleading when taken literally. The clause following that sentence explains what is actually learned. We thank the reviewer for pointing this out and have corrected the phrasing in the revised version of the paper.
>
> ---
>
> # Missing reference to Perona–Malik diffusion
>
> Thank you for the suggestion. We have added the reference in Section 4.1 (Line 297) in the revised version.
>
> ---
>
> # Redundant paraphrased sentence starting in Line 361
>
> We appreciate the reviewer’s comment. The redundancy has been removed in the updated version.
>
> ---
>
> # How is the diffusion coefficient $\sigma$, defined in Eq. 8, used in the instances considered in Sections 5.2 and 5.3?
>
> In the instances considered in Sections 5.2 and 5.3, we set $\lambda_2\equiv\infty$. Thus, in our empirical evaluation we investigated only the specific instance of our general framework (introduced in Section 5.2) in which the drift is anisotropic. The results in Table 1 depend on parameter tuning of $\alpha_1$ and $\lambda_1$. Due to computational resource limitations, we were not able to additionally explore the full generality of the framework with both anisotropic drift *and* anisotropic diffusion.
>
> ---
>
> # There is no proof explaining why the selection of coefficients preserves structures in images
>
> That the diffusion in Eq. 6 is edge-preserving is already established in the classical Perona–Malik work. Beyond this, we provide a “visual justification” in the Design Guidelines in Appendix A. The “transition to isotropy” induced by our parameter choices — which is required for generative modeling (Section 4.3) — is visualized in Appendix E.1.
>
> ---
>
> # Q1: Is there a timeline problem in Line 90?
>
> This question can likely be answered best by the authors of Rissanen et al. (2023) and Hoogeboom & Salimans (2022). Presumably both works went through multiple revision cycles, which may give the impression mentioned by the reviewer.
>
> ---
>
> # Q2: Why are Eqs. (4) and (5) not used anywhere?
>
> From an abstract perspective, the precise form of the loss function is irrelevant for describing the training procedure of a score-based generative model. Since our training setup follows the standard score-based literature and does not differ in this regard from prior work, we did not include an explicit training algorithm in the paper.
>
> # Q3: Why is there a $v$ in Eq. 8, but not in Eq. 6?
>
> The diffusion coefficient $\sigma$ is operator-valued, i.e., for every $(t,u)$ the map $\sigma(t,u)$ is a linear operator from $Q^{1/2}L^2(\Lambda)$ to $L^2(\Lambda)$. The driving Wiener process is $L^2(\Lambda)$-valued. Accordingly, both equations are written in the correct form: Eq. 6 expresses the SPDE in its standard operator notation, while Eq. 8 makes the operator action explicit by writing $\sigma(t,u)v$ for an arbitrary $v$ in the Cameron–Martin space.
>
> ---
>
> # Further writing improvements
>
> In addition to the changes mentioned above, we also revised Table 1 and the caption of Figure 2 to improve clarity. Please refer to the updated version we uploaded.
>
> Thanks,
> The Authors

---

> ### Comment · Reviewer_PX5s · 2025-11-25
>
> I thank the authors taking time to provide the clarification. However, I feel the authors did not address most of my concerns.
> Since the authors introduced the state dependent spde, this makes the solution involving semigroup theory. This result, for example, can be seen in the Figure 1. (3). Further more, Figure 1. (2) shown the pictures with edge preserving. These were corresponding to no noise (as stated in the vertical title), however, the caption stated it as with isotropic noise (a mistake?). Anyway, this example shown in Figure 1. (2)  indeed demonstrates that the proposing algorithm is repeating Perona-Marlik's denoising algorithm to provide the multiscale images while present edges.
>
>  It is unclear how the proposed backward sampling improves the quality of regenerating images. I would appreciate a more rigorous theoretical justification like SBGM for the standard score-based algorithms to support their claims (like Yang Song's DDPM paper).
>
> To my view, the authors failed to show any advantage gained by utilizing Q-winner process and how their backward sampling improves the fidelity of image generation. Namely, the purported advantage of employing the Q-winner process in infinite-dimensional space remains unsubstantiated.
>
> Based on my concerns, I prefer to keep my current score.

---

> > ### Author Response · Authors · 2025-11-25
> >
> > # "Since the authors introduced the state dependent spde, this makes the solution involving semigroup theory. This result, for example, can be seen in the Figure 1. (3)."
> >
> > Unfortunately, we are unable to follow this line of reasoning.
> > While it is true that solutions of certain SPDEs can be formulated using (two-parameter) semigroup theory, we do not see how this observation is connected to Figure 1(3). At present, we are unsure which concrete issue the reviewer seeks to highlight with this remark. We would be happy to address it in detail if clarified.
> >
> > ---
> >
> > # Figure 1(2) shows edge-preserving images corresponding to no noise, but the caption states isotropic noise.
> >
> > Thank you for catching this.
> > The caption was indeed incorrect, and we corrected it in the revised version of the paper.
> >
> > ---
> >
> > # Figure 1(2) demonstrates that the proposed algorithm is simply repeating Perona–Malik’s denoising algorithm.
> >
> > Figure 1 is intended solely to illustrate the **key components** of our forward process:
> >
> > 1. **anisotropic noise**,
> > 2. **anisotropic drift**,
> > 3. **the combination of both**.
> >
> > To isolate these effects, we disabled the drift in (1) and disabled the noise in (2).
> > Thus, (1) indeed reduces to the classical Perona–Malik PDE, which is expected.
> > This is not the forward process used in our generative model — only a didactic visualization of the individual components.
> >
> > ---
> >
> > # It is unclear how the proposed backward sampling improves the quality of regenerated images.
> >
> > First, correctness in the sense that the backward process at terminal time matches the target data distribution is, as in any score-based generative model, guaranteed *by construction* through the formulation of the backward S(P)DE.
> >
> > Second, it is intuitively clear that the longer preservation of geometric structure induced by anisotropy in the forward SPDE should allow the backward sampler to more reliably reconstruct geometric features present in the data.
> >
> > Providing a formal proof of this intuition is challenging and would require a dedicated technical analysis. Ideally, one would aim for upper bounds on metrics such as FID, total variation, or Wasserstein distance expressed in terms of the anisotropy coefficients $\lambda_i$. Developing such theory is an interesting direction for future work.
> >
> > ---
> >
> > # The authors failed to show any advantage gained by utilizing the Q-Wiener process.
> >
> > In our empirical study, our primary focus was on the **anisotropy in the drift**.
> > At the modeling level, however, there was no reason to artificially restrict the SPDE to the special case $Q=\operatorname{id}$. Our framework is general enough to allow arbitrary covariance operators of the driving noise.
> >
> > In practice, this generality comes with a large number of hyperparameters ($\alpha_i$, $\lambda_i$, $\ell$), making a full ablation study prohibitively expensive under submission-period constraints.
> > We therefore restricted attention to the main feature of interest — anisotropy — and demonstrated that this alone already yields substantial improvements in standard generative metrics.
> >
> > ---
> >
> > If there are additional questions, clarifications, or concerns that we can address during the rebuttal period — and that may help improve the reviewer’s overall rating — we kindly ask for a short notification.

---

### Official Review · Reviewer_2Hsy · 2025-10-28

**Soundness:** 2
**Presentation:** 2
**Contribution:** 2
**Rating:** 4
**Confidence:** 3

**Summary:**

This paper proposes a method that employs an anisotropic destruction process, rather than the isotropic noising common in existing diffusion models. Through small-scale experiments, the paper demonstrates performance that is superior or comparable to existing diffusion model families, such as score-based models and flow-matching models.

**Strengths:**

- The paper challenges the convention in the diffusion model literature that we should use an isotropic diffusion process, and it demonstrates the potential of anisotropic SPDEs through comparisons with existing models.

- It clearly explains the conceptual similarities and differences compared to prior work.

**Weaknesses:**

- Standard Gaussian noise-based diffusion models already benefit from schedules that rapidly destroy the image, such as the cosine noise schedule in IDDPM or timestep shifting in Stable Diffusion 3, improving both training and inference.
It is questionable whether the proposed method's superior performance could be achieved in existing score-based or flow-matching models simply by applying more advanced noise schedules.

- The "blurring diffusion models" (Hoogeboom & Salimans), which are only briefly mentioned in the related works, are conceptually very similar as they also use both noise and a blurring drift. A more detailed conceptual and experimental comparison against them is necessary.

**Questions:**

While qualitative comparisons are provided in the appendix, they are limited to CIFAR-10. Can you show qualitative results for ImageNet and LSUN?

---

> ### Author Response · Authors · 2025-11-24
>
> We thank the reviewer for their constructive criticism and suggestions for improvement.
>
> # Can the superior performance of our method also be achieved in existing score-based or flow-matching models by using an appropriate noise schedule?
>
> The so-called *noise schedules* in existing diffusion and flow-matching models correspond most closely to the functions $\alpha_i$ in our SPDE. Modifying these schedules to improve training and/or sampling performance is orthogonal to the performance gains that arise from the *anisotropic destruction* induced directly by our forward process (through its deterministic drift term and stochastic diffusion coefficient).
> In particular, while a noise schedule controls **how strongly** noise is injected over time, the coefficients $\alpha_i$ in our SPDE additionally determine **how this injection interacts with the local image geometry** through gradient-dependent anisotropy—a mechanism that cannot be replicated by a scalar time-dependent noise schedule alone.
>
> ---
>
> # Conceptual similarity to “blurring diffusion models”
>
> The apparent similarity to “blurring diffusion models” is not unique or specific. In fact, regardless of whether one considers OSD, DDPM, SMLD, flow matching, *Generative Modeling with Inverse Heat Dissipation* (Rissanen et al., 2023), or *Score-Based Generative Modeling through SDEs* (Song et al., 2021), the forward processes in all these models can be written in the standard SDE/SPDE decomposition into a **drift term** (deterministic blur/smoothing) and a **diffusion coefficient** (noise injection).
>
> Since the authors of “blurring diffusion models” do not provide code, we were not able—within the limited time of the rebuttal period—to implement their method, train it, and generate samples for a direct empirical comparison. Instead, we must refer to the numbers reported in their paper for any quantitative comparison.
>
> ---
>
> # Qualitive results for ImageNet2012 and LSUN/church_outdoor
>
> We have added qualitative results for ImageNet and LSUN/Church Outdoor in Figures 14 and 17 of the revised version of the paper.

---

> > ### Comment · Reviewer_2Hsy · 2025-11-27
> >
> > Yes, heat dissipation and blurring diffusion models can also be decomposed into a drift term and a diffusion coefficient. However, I do not think this fact by itself implies that the proposed method in this paper is different from, or has novelty over, heat dissipation and blurring diffusion models. Could the authors clarify how this paper differs from these previous works?

---

> > > ### Author Response · Authors · 2025-11-27
> > >
> > > The S(P)DEs considered in *Heat Dissipation* and *Blurring Diffusion Models* correspond only to a very special, degenerate instance of our much more general framework. This exact instance is covered in Section 5.3 of our paper and reduces to a simple stochastic heat equation with **isotropic** noise.
> > >
> > > While our framework is general enough to include this case, such a formulation entirely ignores the central idea behind our approach — namely, introducing **anisotropy** in the drift and/or in the diffusion coefficient.
> > >
> > > The improved metrics we obtain in Section 5.2 arise *precisely because of this anisotropy* in the drift. Such improvements cannot be achieved with the isotropic formulations used in *Heat Dissipation* or *Blurring Diffusion Models*.

---

### Official Review · Reviewer_eGcS · 2025-11-01

**Soundness:** 3
**Presentation:** 2
**Contribution:** 3
**Rating:** 6
**Confidence:** 2

**Summary:**

This paper introduces a novel framework for Score-Based Generative Modeling that
uses anisotropic Stochastic Partial Differential Equations to govern the
diffusion process. The main goal is to enhance image generation quality by preserving the
geometric structure of data during the forward (destruction) process, a departure from
traditional methods that aim to destroy all image information to pure noise. Their forward
process is modeled as the formal solution to SPDE, where it has two components, namely
drift term which enforces deterministic destruction through structural smoothing, and
diffusion term which enables random destruction through noise injection.

**Strengths:**

1). This paper introduces a novel framework that keeps some geometrical structural clues when data destruction, helping the resemblance of geometric features in generative sampling.


2). This paper showcases the proposed method through experiments, and it obtains superior results on both qualitative and quantitative comparisons.

3). One of the main strengths is this unifies formulation of SBGM. i.e., providing a common framework for existing SBGMs and a new anisotropic diffusion process.

**Weaknesses:**

1). I feel even though this paper has the mathematical rigor, it lacks the intuitive and logical building of the proposed method. I suggest authors to add more intuitive explanation that will enable readers to understand the paper much better. Otherwise, the current version is
bit hard to follow and grasp the concepts. If possible, try to add a figure that explaining the concept.


2). The experimental section only compares against methods up to 2023. For completeness, I suggest including comparisons or discussions of more recent works such as “Edge-Preserving Noise for Diffusion Models” (2024), which shares a similar motivation of geometry-aware corruption.

3). The proposed method requires almost 2000 score evolutions. This I feel a major limitation compared to recent works.

4). The authors mainly used \( \ell = 0 \) for the noise process. I would like to know whether introducing a finite \( \ell > 0 \) — i.e., spatially correlated noise could help capture textured patterns or reduce artifacts, or if it would mainly complicate the sampling
process.

5). In this paper, both the drift and diffusion terms depend on the local gradient. I would like to know whether it is possible to understand how sensitive the model’s performance is to this gradient dependence. For instance, if ( g_1 ) varies too sharply with ( \nabla u ), could it lead to unstable training dynamics? I feel some empirical or theoretical insight into this behavior would be useful.


6). The core idea centers on the argument of a “residual dependence on the initial image.” I would like to know whether the authors attempted to measure how much information about the initial image remains at t=T. Can this dependence be quantified using a specific metric or statistical measure? I believe such an analysis would provide deeper insight into the behavior and effectiveness of the proposed method.

**Questions:**

see the weaknesses section.

---

> ### Author Response · Authors · 2025-11-24
>
> We thank the reviewer for the thoughtful and constructive feedback.
>
> # W1: Can we add an illustrative figure that illustrates the concept?
> We appreciate the concern. We would like to draw attention to our **Appendix A (Design Guideline)**, where we give intuitive explanations and visualizations that describe each component of our anisotropic SPDE and their effect on the forward destruction process. Beyond the specifics of our SPDE as the forward process, the methodology of our framework is the same as in every score-based model.
>
> # W2: Can we compare to more recent works like Vandersanden et al. (2024)?
> Thank you for this suggestion. We have added a quantitative comparison to *Edge-Preserving Noise for Diffusion Models* (Vandersanden et al., 2024) in **Table 1** of the revised paper.
>
> # W3: Requiring almost 2000 score evaluations seems to be a major limitation
> The total number of score evaluations depends strongly on the chosen time discretization and, in particular, on how conservative the time-stepping scheme is. In our empirical study, we deliberately selected a **very small step size** ($\Delta t=0.001$) to guarantee maximal numerical stability and highest sample quality. Less conservative step sizes are certainly possible and would substantially reduce the number of required score evaluations, but exploring this trade-off was beyond the scope of the present work and is left for future investigation.
>
> # W4: Does introducing a strictly positive $\ell>0$ help improving the results?
> We chose $\ell=0$ to isolate the effect of the anisotropic drift term and avoid confounding it with additional structure injected through the diffusion coefficient. Introducing spatially correlated noise ($\ell>0$) would require a completely new set of hyperparameter searches and additional compute. This is an interesting direction for future work. Due to hardware resource limitations a full abliation study for the hyperparameters was out-of-scope for this submission.
>
> # W5: Does the choice of $g_i$ influence stability of the (numerical) simulation?
> Yes. If $g_i$ changes too sharply as a function of the spatial gradient $\nabla u$, the numerical simulation can indeed become unstable. This behavior is well-known in the anisotropic diffusion literature. For a theoretical analysis of such stability issues, we refer to the study:
>
> **Selim Esedoglu**, *Stability Properties of the Perona–Malik Scheme*,
> https://doi.org/10.1137/S0036142903424817.
>
> In short, highly sensitive or discontinuous choices of $g_i$ may lead to ill-posedness or numerical blow-up, whereas smooth and gently varying anisotropy coefficients yield stable and well-behaved evolutions.
>
> # W6: Can we measure theoretically the residual dependence on the initial data?
> Yes, in principle this can be quantified. However, doing so requires a highly technical analytical investigation of the underlying SPDE, which is beyond the scope of this submission. Standard references in the SPDE literature, e.g.
>
> **Giuseppe Da Prato & Jerzy Zabczyk**, *Stochastic Equations in Infinite Dimensions*,
> https://doi.org/10.1017/CBO9780511666223,
>
> already contain results on the dependence of SPDE solutions on initial data for more general equations than those considered here. These results can be adapted to our setting, but carrying this out rigorously would require a dedicated theoretical treatment.

---

### Official Review · Reviewer_HKrC · 2025-11-03

**Soundness:** 3
**Presentation:** 3
**Contribution:** 3
**Rating:** 4
**Confidence:** 3

**Summary:**

This paper proposes an extension of score-based generative models (SBGMs) by formulating the forward diffusion process using anisotropic Stochastic Partial Differential Equations (SPDEs) instead of the more common Stochastic Differential Equations (SDEs). The authors argue that traditional SDE-based methods, which are typically isotropic, destroy all information uniformly, including valuable geometric structures.

The core contribution is the introduction of nonlinear, spatially-dependent SPDEs where the drift and diffusion coefficients are influenced by anisotropy coefficients. The goal with this formulation is to more generally preserve spatially relevant information such as edges during the forward process, so that the reverse process can in turn leverage these geometric cues to sample images with higher fidelity.

**Strengths:**

I believe the paper's formulation is novel and interesting. Some strengths are:

**(S1)**: Novel theoretical contribution. A general framework for spatially-dependent diffusion processes is quite valuable as it can capture more complex dynamics in structured data like images. I particularly like how many different forms of data corruption in the forward process are subsumed under the same framework. I would like to see this extended further across modalities.

**(S2)**: Experimental validation on pre-trained model. The authors demonstrate that their training on top of an existing diffusion model yields improvements in standard image generation quality metrics such as FID and inception score.

**(S3)**: Clear presentation for the theoretical section. I appreciate the care with which the authors explained the differences in diffusion frameworks and then unified it under one umbrella. This set up the motivation and intuition for the core method well-- preserving geometric structures via anisotropy will aid in reconstruction.

Overall, I think the idea is interesting and deserves further exploration.

**Weaknesses:**

**(W1)**: Lack of thorough experimental validation. This is my main concern with the paper. There are a number of baselines and prior relevant work that have not been included in the experimental results (eg: Table 1). This makes it hard to judge the efficacy of the method. E.g. [1] achieved an FID of 1.97 on CIFAR-10. [2] tackles a similar problem as this paper, but results have not been compared.

**(W2)**: Computational cost. The forward and backward process requires the computation of spatial gradients (eq 7, 8) via finite differences, for every time step. This would arguably make training and inference much slower. A comparison between the quality / performance tradeoff with prior work and baselines is critical and is missing.

**(W3)**: Extension to latent diffusion models. While the theoretical framework should apply equally to standard and latent diffusion models, I would be curious to see empirical results on latent diffusion models, which have become the standard today. Do the findings still hold? This is important for broader relevance and applicability.

**(W4)**: While the authors do discuss the anisotropy, diffusivity and intensity coefficients in Appendix A, experimental validation for different choices of hyperparameters is missing. There are multiple new hyperparameters introduced in this paper and a detailed ablation would be quite important.

The weaknesses slightly outweigh the strengths for me. I would encourage the authors to present more extensive empirical results to support their claims.

---
References:
[1] Elucidating the Design Space of Diffusion-Based Generative Models, NeurIPS 2022.
[2] Edge-preserving noise for diffusion models, arXiv 2410.01540.

**Questions:**

**(Q1)**: L456. "Notably, according to the original authors, continuing training with their own method did not yield further metric improvements."
Was this verified via experimental results on your side?

**(Q2)**: Table 1. Why is the Ours (Isotropic) version so much worse on FID than the anisotropic version? What are the results on the remaining datasets?

**(Q3)**: Are there any results on the fully anisotropic variant? Where both the diffusion and drift terms are anisotropic.

**(Q4)**: The fine-tuning experiment (Figure 2)  is interesting. Does this imply that the primary benefit is in the sampling path (i.e., the backward SDE derived from the anisotropic SPDE is simply a better path from noise to data), or is the score model itself being fundamentally retrained to leverage geometric information that it was previously ignoring?

**(Q5)**: The exposition on different diffusion methods was clear and well-written. Section 4.1 was a bit opaque to me. It was a bit difficult for me to get an intuitive sense of the terms in eqs. 7, 8, 9. Some additional explanation would be valuable for this section.

---

> ### Author Response · Authors · 2025-11-24
>
> We thank the reviewer for their careful reading of our work and their constructive and insightful comments. Below we address each point raised.
>
> ---
>
> # **W1: Comparison to *Edge-Preserving Noise for Diffusion Models* (Vandersanden et al., 2024)**
>
> A quantitative comparison to Vandersanden et al. (2024) has been included in Table 1 of the revised paper.
>
> ---
>
> # W2: Computational costs
>
> To contextualize our computational costs relative to Song et al. (2021), we report below the **normalized training times per 10k steps** on CelebA (64):
>
> | Model                | Time / 10k steps |
> |----------------------|------------------|
> | Song et al. (2021)   | 2506.55 s    |
> | Ours (anisotropic)   | 4658.63 s    |
>
> ---
>
> # **W3: Extension to latent diffusion models**
>
> We appreciate the reviewer raising this important point.
> Conceptually, our framework applies equally well to pixel-space and latent-space diffusion models:
> the forward SPDE acts on an abstract function space, and the derivation of the reverse-time SDE/SPSDE is agnostic to whether the underlying representation corresponds to images or latent representations produced by an autoencoder.
>
> However, conducting a proper empirical evaluation in latent space requires:
>
> 1. **Complete retraining** of a latent diffusion model from scratch,
> 2. **Re-tuning** all anisotropy coefficients (\(\alpha_i, \lambda_i\)) to match the statistics and geometry of the latent manifold.
>
> This constitutes a substantial new experimental pipeline that probably cannot reasonably be executed during the rebuttal period. If we obtain preliminary results during the discussion period, we will upload them as an additional comment.
>
> We fully agree that applying anisotropic SPDEs in latent space is an exciting and relevant research direction and plan to pursue this in future work.
>
> ---
>
> # **W4: Ablation study for hyperparameters**
>
> We agree with the reviewer that an ablation study over the hyperparameters of our SPDE would be insightful, even for the specific instance evaluated in Section 5.2, where only $\alpha_1$ and $\lambda_1$ are active.
>
> However, each such experiment requires multiple full trainings and large-scale sampling, which is computationally expensive. Given our hardware resource limitations, we prioritized demonstrating the core phenomenon rather than exhaustively exploring the hyperparameter space.
>
> ---
>
> # **Q1: Verification that further training of Song et al. (2021) does not improve their baseline**
>
> We did not re-train Song et al. (2021).
> The authors explicitly state that the publicly released checkpoint is the one that achieved their best reported results. To rigorously verify this claim, one would need to train  Song et al. (2021) from scratch over multiple training schedules, which is outside the scope of our work and orthogonal to the scientific contribution we make.
>
> ---
>
> # **Q2: Why is *Ours (isotropic)* substantially weaker, and how does it perform on other datasets?**
>
> The *Ours (isotropic)* baseline corresponds to a **stochastic heat equation with additive noise**, which is a special — and significantly degenerate — instance of our framework.
> We included this case primarily to demonstrate that our anisotropic SPDE subsumes the “inverse heat dissipation” forward process used in *Rissanen et al. (2023)*, while showing that our score-based reverse process achieves superior results than theirs.
>
> Given the much poorer CIFAR-10 performance, we do not expect competitive results on higher-resolution datasets, and therefore did not conduct additional experiments. Importantly, this isotropic baseline fully disables anisotropy in both drift and diffusion and thus cannot exhibit the structural advantages our method is designed to capture.
>
> ---
>
> # **Q3: Results for an SPDE with *both* anisotropic drift and anisotropic diffusion**
>
> No. In this work we investigated only the instance presented in Section 5.2, which features an anisotropic drift term. Extending this to a double-anisotropic configuration (anisotropic drift and anisotropic diffusion) requires extensive parameter tuning ($\alpha_i$, $\lambda_i$) and full training runs, which we could not perform due to resource limitations.
>
> ---
>
> # **Q4: Does Figure 2 indicate that the main benefit of our method lies in the sampling phase?**
>
> No.
> Figure 2 demonstrates that **even a pre-trained model** can be **significantly improved** with only a small number of additional training iterations under our SPDE framework.
> The simulation scheme introduced in Eq. 39 — used for both forward and reverse S(P)DEs — is primarily for **numerical stability**, not the source of the improved generative performance.
>
> ---
>
> # **Q5: Intuitive explanation of the SPDE components**
>
> Appendix A contains a detailed design guideline that explains the intuitive role of each term in our SPDE and provides visualizations illustrating their effects.

---

> ### Comment · Reviewer_HKrC · 2025-11-24
> **Response to Rebuttal**
>
> I thank the authors for their response. As a word of advice, I would suggest that any revisions to the main text of the paper be highlighted in a different color (eg: red or blue text) so it is easier for reviewers to track changes.
>
> **(W1)**: The FID scores for Vandersanden et al. seems quite high. For example, they reported 23.17 FID on LSUN 128x128, whereas the result in Table 1 is 49.1 for LSUN 256x256. Please explain the discrepancy and describe the experimental configuration used to reproduce their results. Further, there are other baselines that have still not been added to Table 1.
>
> **(W2)**: What is the inference cost? I also don't see this training cost table added to the revised paper. The lack of discussion around computational cost remains a significant weakness. Also, this remains unaddressed:
> > A comparison between the quality / performance tradeoff with prior work and baselines is critical and is missing.
>
>
> **(W3)**: I understand the compute limitations, but lack of experimentation on latent diffusion models remains a weakness for me. Also- why does the LDM need to be retrained from scratch? In your experiments in the paper for pixel-space diffusion models, you started with a pre-trained checkpoint from Song et al. (2021) and then further trained with your framework. Can the same not be applied here?
>
> **(W4)**: Some form of ablation study is quite important for readers and future users of your work to understand the tradeoffs of the method you are proposing. I did not ask for an exhaustive exploration.
>
> **(Q1)**: My main point here was to establish a valid "control" for your results in Table 1. Presenting the results of Song et al. from their trained checkpoint is good, but a valid control experiment to contrast your SPDE is to continue training the pre-trained checkpoint under the original diffusion SDE for as many training steps as you trained your model with the SPDE.

---

> ### Author Response · Authors · 2025-11-25
>
> We thank the reviewer for the thoughtful feedback. All changes relative to the original submission are now highlighted with yellow boxes in the revised paper.
>
> # W1a: Results for Vandersanden et al. (2024)
> We obtained the generated samples for Vandersanden et al. (2024) directly from the authors. The higher FID scores on LSUN at 256×256 (compared to 128×128) appear to stem from the fact that their method empirically performs worse at higher resolutions.
>
> # W1b: Other baselines
> Unfortunately, it is not feasible to integrate additional baselines into Table 1 during the rebuttal period. A further complication is that several highly relevant baselines — such as *Blurring Diffusion Models* (Hoogeboom & Salimans) — do not provide publicly available code.
> A careful inspection of related work also shows that many papers simply quote the numbers reported in the baseline papers rather than re-training those models. Since absolute values of metrics such as FID can depend sensitively on implementation details, we consider it inappropriate to include such numbers “literally” in Table 1. For  *Blurring Diffusion Models* (Hoogeboom & Salimans), the authors report an FID score for CIFAR-10 of 3.17 which is worse than Ours (anisotropic). Other numbers relevant to our experiments are not reported in *Blurring Diffusion Models* (Hoogeboom & Salimans).
>
> # W2a: Inference cost
> We added a section *Computational costs* in Appendix N, which now reports both training and inference costs. As the timings shown there make clear, our method achieves significantly improved inference times compared to Song et al. (2021), due to our numerical implementation explained in Appendix H — despite having a theoretically more demanding drift and diffusion coefficient.
>
> # W2b: Tradeoff between performance and computational cost
> The desired quality/performance tradeoff can now be directly assessed by combining the cost numbers in Appendix N with the quantitative metrics in Table 1, especially with respect to the key baseline of Song et al. (2021), which marks the transition from SDEs to our SPDE framework.
>
> # W3: Can we initialize latent-space training from a pre-trained checkpoint?
> Yes, in principle training in latent space could also be initialized from a pre-trained checkpoint. However, for the datasets considered in our work, no such checkpoint exists. Hence, a full latent-space evaluation would require complete re-training from scratch together with parameter re-tuning.
>
> # W4: Ablation study on hyperparameters
> We fully agree that studying the sensitivity of the results with respect to the hyperparameters would be highly desirable. Our “Ours (isotropic)” baseline already illustrates the effect of varying $\lambda_1$, since this instance corresponds to the degenerate case obtained by transitioning from $\lambda_1=\infty$ to a schedule from $\lambda_1=0.025$ to $\lambda_1=\infty$.
> Given the large number of degrees of freedom in the full SPDE framework ($\alpha_i$, $\lambda_i$, $\ell$), an exhaustive ablation would require substantial additional compute.
> In our view, the most meaningful future direction would be a *theoretical* analysis yielding upper bounds on standard metrics (FID, TV distance, Wasserstein distance, …) in terms of the hyperparameters. Such an analysis is beyond the scope of this submission and would merit an independent theoretical study.
>
> # Q1: Clarification of the results in Table 1
> For clarity: **all results in Table 1 correspond to trainings from scratch** for both Song et al. (2021) and our method.
> Only Figure 2 shows results obtained by fine-tuning a pre-trained checkpoint.
>
> The checkpoint of Song et al. (2021) used for Figure 2 is the publicly released one, which the authors themselves report as achieving their best quantitative performance. According to their paper, additional training iterations beyond this checkpoint degrade the metrics, so re-training or extending their training was outside the scope of our study.
>
> We hope these clarifications help resolve the reviewer’s concerns.

---

> > ### Author Response · Authors · 2025-11-27
> >
> > # W3: Latent space experiment
> >
> > A latent-space experiment for LSUN is currently underway, and we are confident that we can share results before the end of the discussion period.

---

> ### Author Response · Authors · 2025-12-01
>
> # Latent Space Experiment
>
> We conducted a latent space experiment using *Ours (anisotropic)* on LSUN/church_outdoor (256x256).
> For the latent representation, we used the pretrained VAE **stabilityai/sd-vae-ft-mse** (via `FlaxAutoencoderKL`).
>
> **Results:**
> - Inception Score: **3.880973**
> - FID: **3.936322**
> - KID: **2.387085e-03**
>
> These results demonstrate that our anisotropic SPDE framework also performs well in latent space and even achieves superior results.

---

### Author Response · Authors · 2025-11-25

We thank the reviewers for the extensive and constructive feedback. Below we summarize the key revisions made to improve the paper.

# Major Revisions and Additions

- We improved/enhanced the captions of **Figure 2** and **Table 1**.
- We added a new **Computational Cost** section in **Appendix N**, reporting both training and inference times.
- We replaced **Fig. 11** with updated CelebA results for **Ours (anisotropic)**.
- We now include **uncurated generated samples** across datasets:
  - **Appendix K (Fig. 11–13):** CelebA — *Ours (anisotropic)*, Song et al. (2021), Lipman et al. (2023).
  - **Appendix L (Fig. 14–16):** ImageNet2012 — *Ours (anisotropic)*, Song et al. (2021), Lipman et al. (2023).
  - **Appendix M (Fig. 17–18):** LSUN/church_outdoor — *Ours (anisotropic)* and Song et al. (2021).

# Latent Space Experiment

We additionally performed a latent–space experiment on LSUN/church\_outdoor (256×256) using **Ours (anisotropic)**.
For the latent representation, we employed the pretrained VAE **stabilityai/sd-vae-ft-mse** (via `FlaxAutoencoderKL`). We compared our results to Song et al. (2021):

| Model                       | IS ↑     | FID ↓     | KID ↓        |
|-----------------------------|----------|-----------|--------------|
| **Ours (anisotropic)** | **3.880973** | **3.936322** | **2.387085e-3** |
| *Song et al. (2021)*               | 2.1550   | 13.372    | 7.473e-3     |

These results demonstrate that our anisotropic SPDE framework also performs strongly in latent space.
The experiment and corresponding uncurated samples are now included in **Appendix O** and **Appendix P**.

# Computational Costs

Appendix N now provides a clear comparison of **training** and **inference** costs.
As shown, our method requires only slightly more training time—despite the theoretically increased complexity — due to our efficient implementation (Appendix H).
Notably, the **inference time is significantly reduced**.

# Additional Improvements

- We extended **Table 1** with quantitative results from the recent anisotropic approach of **Vandersanden et al. (2024)**.
- All writing suggestions and missing references pointed out by the reviewers have been incorporated.

---

### Author Response · Authors · 2025-12-01
**To the New AC(s)**

We would like to sincerely thank the new AC(s) for engaging with our work and for their efforts. We believe that we have provided a strong rebuttal that addresses and resolves the remaining ambiguities and weaknesses raised by the reviewers, thereby significantly improving the paper.

Nonetheless, we would greatly appreciate receiving an indication in the near future of how the new AC(s) assess our rebuttal, so that we can provide any additional responses and/or results if necessary.

---

### Note · Authors · 2026-01-26

I have read and agree with the venue's withdrawal policy on behalf of myself and my co-authors.

---

### Meta-Review · Area_Chair_GoFu · 2026-01-07

**Summary:**

This paper proposes an *anisotropic* stochastic partial differential equation (SPDE) framework for score-based generative modeling, to preserve geometric structure during the forward diffusion process. Reviewers generally agree that the paper presents an interesting and theoretically motivated extension of existing diffusion frameworks, and several of them appreciate the unifying formulation and the empirical improvements.

However, the reviews consistently raised concerns regarding the strength and completeness of the empirical validation. Specifically, the novelty of the proposed framework relative to existing anisotropic or blurring-based diffusion models, and the lack of systematic analysis of computational tradeoffs and hyperparameters. While the rebuttal addressed some issues by adding latent-space experiments, additional qualitative results, and clarifications on numerical stability and backward sampling correctness, several core concerns remain unresolved.

In particular, reviewers questioned whether the empirical improvements can be clearly attributed to the proposed anisotropic SPDE formulation, given the incomplete baseline coverage, unclear alignment with recent related methods, and absence of ablation studies. As a result, the paper was generally viewed as promising but not yet sufficiently validated to meet the acceptance bar.

**Reviewer Concerns:**

## Addressed

**Latent diffusion model (HKrC)**
- **Concern:** The generalization to the latent diffusion model
- **Action:** Authors added a latent-space experiment on LSUN/church_outdoor using a pretrained VAE, reporting promising IS/FID/KID metrics.

**Limited qualitative results (2Hsy)**
- **Concern:** Qualitative comparisons were limited to CIFAR-10.
- **Action:** Authors added qualitative samples for ImageNet2012 and LSUN/church_outdoor in the revised paper.

**Stability (eGcS)**
- **Concern:** Sharp dependence on image gradients could lead to unstable dynamics.
- **Action:** Authors provided theoretical discussion and references (e.g., Perona-Malik stability analysis) explaining when stability holds.

**Backward sampling (PX5s)**
- **Concern:** Whether the backward process is well-defined and recovers the target distribution.
- **Action:** Authors clarified that standard SDE/SPDE time-reversal theory applies after finite-dimensional discretization.
- **Status:** **Addressed at the correctness level** (but not novelty)

**Writing (PX5s, eGcS)**
- **Concern:** Poor flow, missing classical references, redundant text, unclear captions.
- **Action:** Authors revised wording, added references (e.g., Perona-Malik), corrected captions, and removed redundancies.

**Explanation and interpretability (eGcS)**
- **Concern:** The method is mathematically heavy and difficult to intuitively understand.
- **Action:** Authors referred to Appendix A for intuition and visualizations.
- **Status:** **Partially addressed**


---

## Still outstanding

**Baseline comparisons (HKrC, eGcS)
- **Concern:** Table 1 lacks several important baselines;
- **Action:** Authors added Vandersanden et al. results, however, the added comparison to Vandersanden et al. (2024) is not exact aligned with the paper.
- **Status:** Reviewer explicitly questioned the alignment of experimental configurations and noted that other baselines remain missing.


**Cost-quality tradeoff (HKrC)**
- **Concern:** Gradient-based SPDEs may incur high cost; no explicit tradeoff analysis.
- **Action:** Authors added training and inference costs in Appendix N.
- **Status:** Reviewer indicated the tradeoff is still not explicitly analyzed across baselines.


**Hyperparameter analysis (HKrC, eGcS)**
- **Concern:** Many new anisotropy-related hyperparameters were introduced without ablation.
- **Action:** Authors argued ablations are computationally expensive and deferred to future work.

**High sampling cost (eGcS)
- **Concern:** Nearly 2000 score evaluations are expensive compared to recent methods.
- **Action:** Authors stated conservative step size was chosen for stability; faster settings not explored.
- **Status:** AC suggests that the authors compare their method with others under varying sampling steps.

**Novelty (2Hsy, PX5s)**
- **Concern:** Proposed method may be a reformulation of existing anisotropic diffusion (e.g., Perona-Malik, blurring diffusion, Vandersanden et al.).
- **Action:** Authors argued these are degenerate isotropic cases of their proposed unified framework. However, reviewers remained unconvinced and explicitly indicated that they would maintain their scores. The authors further explained that the proposed novelty lies in introducing anisotropy in the drift and/or in the diffusion coefficient.
- **Status:** This explanation didn't get a response. AC checked the paper and didn't find the empirical results to show the difference.

**Central claim not empirically or theoretically quantified (eGcS)**
- **Concern:** Lack of a quantitative measure of “residual dependence on initial data.
- **Action:** Authors acknowledged the importance but deferred to future theoretical work.

**Lack of theoretical justification (PX5s)**
- **Concern:** No rigorous theory shows that anisotropic SPDE improves generation quality beyond classical diffusion.
- **Action:** Authors provided intuition and correctness arguments, but no formal proof.  They further explain that the experiments are infeasible.

**Reviewer Scores:**

**Reviewer HKrC (4 $\rightarrow$ 4):**
The rebuttal addressed the latent diffusion concern and added computational cost analysis, but additional baseline comparisons and cost-quality tradeoff analysis remain incomplete, so the score would likely stay the same or increase only slightly.

**Reviewer eGcS (6 $\rightarrow$ 6):**
Appreciated the theoretical framework but questioned interpretability, sampling efficiency, lack of ablations, and missing quantitative validation of core claims. Some issues were partially addressed, but high sampling cost and limited empirical grounding remain.

**Reviewer 2Hsy (4 $\rightarrow$ 4):**
Questioned novelty relative to blurring and heat-dissipation diffusion models. Despite conceptual clarifications, there is no empirical evidence.

**Reviewer PX5s (2 $\rightarrow$ 2):**
Raised strong concerns about theoretical justification, novelty, and empirical evidence. The reviewer explicitly stated that these concerns were not resolved and maintained the rejection score.

---

### Decision · Program_Chairs · 2026-01-26

Reject